# Overview and statistical analysis of boundary layer clouds and precipitation over the western North-Atlantic Ocean

Simon Kirschler[1,2], Christiane Voigt[1,2], Bruce E. Anderson[3], Gao Chen[3], Ewan C. Crosbie[3], Richard A. Ferrare[3], Valerian Hahn[1,2], Johnathan W. Hair[3], Stefan Kaufmann[1,2], Richard H. Moore[3], David Painemal[3], Claire E. Robinson[3], Kevin J. Sanchez[3], Amy J. Scarino[3], Taylor J. Shingler[3], Michael A. Shook[3], Kenneth L. Thornhill[3], Edward L. Winstead[3], Luke D. Ziemba[3], and Armin Sorooshian[4,5]

[1]Institut für Physik der Atmosphäre, Deutsches Zentrum für Luft- und Raumfahrt (DLR), Oberpfaffenhofen, Germany
[2]Institut für Physik der Atmosphäre, Johannes Gutenberg-Universität, Mainz, Germany
[3]NASA Langley Research Center, Hampton, VA, USA
[4]Department of Chemical and Environmental Engineering, University of Arizona, Tucson, Arizona, USA
[5]Department of Hydrology and Atmospheric Sciences, University of Arizona, Tucson, Arizona, USA

**Correspondence:** Kirschler Simon (Simon.Kirschler@dlr.de)

**Abstract.** Due to their fast evolution and large natural variability in macro- and microphysical properties, the accurate representation of boundary layer clouds in current climate models remains a challenge. One of the regions with large intermodel spread of the Coupled Model Intercomparison Project Phase 6 ensemble is the western North-Atlantic Ocean. Here, statistically representative in-situ measurements can help to develop and constrain the parameterization of clouds in global models. To this end, we performed comprehensive measurements of boundary layer clouds, aerosol, trace gases, and radiation in the western North-Atlantic Ocean during the NASA Aerosol Cloud meTeorology Interactions oVer the western ATlantic Experiment (AC-TIVATE) mission. 174 research flights with 574 flight hours for cloud and precipitation measurements were performed with the HU-25 Falcon during three winter (February-March 2020, January-April 2021, and November 2021-March 2022) and three summer seasons (August-September 2020, May-June 2021, and May-June 2022). Here we present a statistical evaluation of 17209 individual cloud events probed by the Fast Cloud Droplet Probe and the Two-Dimensional Stereo cloud probe during 155 research flights in a representative and repetitive flight strategy allowing for robust statistical data analyses. We show that the vertical profiles of distributions of the liquid water content and the cloud droplet effective diameter (ED) increase with altitude in the marine boundary layer. Due to higher updraft speeds, higher cloud droplet number concentrations ($N_{liquid}$) were measured in winter compared to summer despite lower cloud condensation nuclei abundance. Flight cloud cover derived from statistical analysis of in-situ data is reduced in summer and shows large variability. This seasonal contrast in cloud coverage is consistent with a dominance of a synoptic pattern in winter that favors conditions for the formation of stratiform clouds in the western edge of cyclones (post-cyclonic). In contrast, a dominant summer anticyclone is concomitant with the occurrence of shallow cumulus clouds and lower cloud coverage. The evaluation of boundary layer clouds and precipitation in the $N_{liquid}$-ED phase space sheds light on liquid, mixed-phase, and ice cloud properties and helps to categorize the cloud data. Ice and liquid precipitation, often masked in cloud statistics by high abundance of liquid clouds, is often observed throughout the cloud. The ACTIVATE in-situ cloud measurements provide a wealth of cloud information useful for assessing airborne and satellite

remote sensing products, for global climate and weather model evaluations, and for dedicated process studies that address precipitation and aerosol-cloud interactions.

## 1 Introduction

Low-level clouds play a significant role in the climate system. They reflect shortwave solar radiation and prevent it from reaching the Earth's surface, which leads to cooling (Hartmann et al., 1992; Stephens et al., 2012; Henderson et al., 2013; Gettelman and Sherwood, 2016). As temperatures of low-level clouds are close to surface temperatures, the absorption of terrestrial longwave radiation by low-level clouds has a minor warming effect (IPCC, 2013). The large negative shortwave cloud radiative cooling effect together with a negligible longwave cloud radiative warming effect results in higher cooling rates by low-level clouds compared to other cloud types (Hartmann et al., 1992; Wang et al., 2023). Weather systems affect cloud cover and microphysical properties and can induce ice nucleation or the formation of precipitation (Painemal et al., 2023; Naud and Kahn, 2015), which in turn affects their radiative properties. Hence, due to the fast evolution and large natural variability of clouds, the representation of clouds in climate models remains a challenge (Mülmenstädt and Feingold, 2018). The multimodel net cloud feedback in Coupled Model Intercomparison Project Phase 5 (CMIP5) models ranges from $-0.13$ to $1.24$ W m$^{-2}$ K$^{-1}$ (Ceppi et al., 2017) and shows a larger range in CMIP6 models with an increase in their mean values from $0.09$ to $0.21$ W m$^{-2}$ K$^{-1}$ due to a decrease in low cloud coverage (Zelinka et al., 2020). Cesana et al. (2022) show that the reflected shortwave solar radiation is still underestimated in CMIP6 models compared to satellite observations in the Southern Ocean. This calls for the need of constraining cloud parameterizations in global climate models with observational data (e.g. IPCC, 2021). Marine low-level clouds cover more than 45% of the ocean surface (Warren et al., 1988). Tselioudis et al. (2013) show that in particular the occurrence of low-level broken or shallow cloud systems are significantly underestimated by the CMIP5 models. In the following CMIP6, global models improved their cloud parameterizations and simulated a stronger shortwave cloud feedback compared to CMIP5 model versions; however, CMIP6 shows larger intermodel spread in effective climate sensitivity than CMIP5 (Bock et al., 2020).

The western North-Atlantic Ocean (WNAO) is one of the regions where the CMIP6 multi-model mean surface temperature significantly departs from observations (Bock et al., 2020). The WNAO has a broad spectrum of aerosol sources, species and abundances (Sorooshian et al., 2020; Corral et al., 2021). In addition, the WNAO has a wide range of meteorological conditions with mainly low shallow cumulus clouds and episodic occurrence of marine stratocumulus clouds (Tselioudis et al., 2013) and frontal systems (Field et al., 2017a). This provides ideal conditions for assessing the representation of cloud and aerosols in climate models. In-situ cloud obervations have been used to evaluate and constrain large eddy simulation (LES) in cold air outbreak (CAO) situations in the WNAO and are able to reproduce the marine boundary layer (MBL) meteorology (Li et al., 2022, 2023). Other LES studies point to the importance of mixed-phase cloud processes and the need of in-situ observations for evaluation. Other studies found that riming, the collection of droplets, and consequential reduction of aerosol concentration, promotes the transition from overcast to broken clouds in the WNAO (Tornow et al., 2021; Goren et al., 2022; Abel et al., 2017).

Additionally, entrainment from the free troposphere dilutes the aerosol concentration downwind offshore and accelerates this transition (Tornow et al., 2022; Wood et al., 2011).

The WNAO MBL is influenced by the Gulf Stream and the Bermuda-Azores anticyclone in summer (Sorooshian et al., 2020). The anticyclone drives south-westerly winds near sea surface, reaching their maximum in summer. It moves south-westward during winter primarily due to the development of a cyclonic low north of 45°N, which promotes strong north-westerly winds. The north-westerly winds are accompanied with strong sensible heat fluxes and vertical motions during winter (Painemal et al., 2021) leading to supersaturation of moist air and cloud formation by droplet activation of the available cloud condensation nuclei. In the life cycle of clouds, after cloud droplet activation, further impact and growth processes of cloud particles take place, which change the microphysical properties of the cloud. Immediately after cloud droplet activation, condensation is the most important growth process up to droplet diameters of about $15\ \mu m$ (Pruppacher and Klett, 2010). The thermodynamic system of a cloud strives to reduce the present supersaturation. Maximizing the surface area of the water-water vapor interface layer serves for an effective reduction by condensation. Therefore, higher cloud droplet number concentrations ($N_{liquid}$) lead to faster reduction of supersaturation and smaller cloud droplets due to water vapor competition (Pinsky et al., 2012). Another important growth process of liquid clouds is collision-coalescence (Pruppacher and Klett, 2010). Since cloud droplets $<10\ \mu m$ have very low collision probabilities with other cloud droplets (Böhm, 1992b), high concentrations of cloud condensation nuclei can suppress the formation of precipitation through reduced coalescence (Albrecht, 1989; Freud and Rosenfeld, 2012; Braga et al., 2021b). The growth of ice particles by condensation works analogously to that of cloud droplets. If the air is superaturated with respect to ice and undersaturated with respect to liquid in mixed-phase clouds, the growth of ice particles by condensation is significantly accelerated as the water droplets evaporate and act as an additional water vapor reservoir, this is called the Wegener-Bergeron-Findeisen process (WBF) (Wegener, 1911; Bergeron, 1935; Findeisen, 1938; Korolev, 2007b, 2008). Riming describes ice growth by the accumulation of supercooled water droplets that freeze on contact with ice particles, leading to the glaciation of mixed-phase clouds (Böhm, 1992a; Pruppacher and Klett, 2010). The glaciation of clouds is further accelerated by secondary ice production processes that increase the ice number concentration (Hallett and Mossop, 1974; Field et al., 2017b; Korolev et al., 2020; Keinert et al., 2020; Luke et al., 2021; Lawson et al., 2023). Overall, dynamics and aerosols are key parameters for cloud formation. Recent studies show an aerosol gradient over the WNAO with lower number concentrations and aerosol optical depth off the coast and strong aerosol transport from the continent (Corral et al., 2021). In addition, variables representing aerosol abundance (e.g., aerosol optical depth, number concentration) over this region show a marked annual cycle, with minimum and maximum values in winter and summer, respectively (Dadashazar et al., 2021b). Despite high cloud condensation nuclei number concentrations in summer, the dynamical impact of updraft speeds dominates the cloud formation process and leads to higher $N_{liquid}$ in winter (Kirschler et al., 2022). Altogether the WNAO features interesting and complex weather patterns, providing a natural laboratory for studying liquid and mixed-phase shallow and broken cumulus clouds in a broad spectrum of aerosol and meteorological conditions.

Here, we present an overview of microphysical properties of low-level clouds measured during the Aerosol Cloud meTe-orology Interactions oVer the western ATlantic Experiment (ACTIVATE) campaign (Sorooshian et al., 2019). We investigate the spatial, seasonal and altitude dependence of $N_{liquid}$, ice number concentrations ($N_{ice}$), effective diameter of liquid ($ED_{liquid}$)

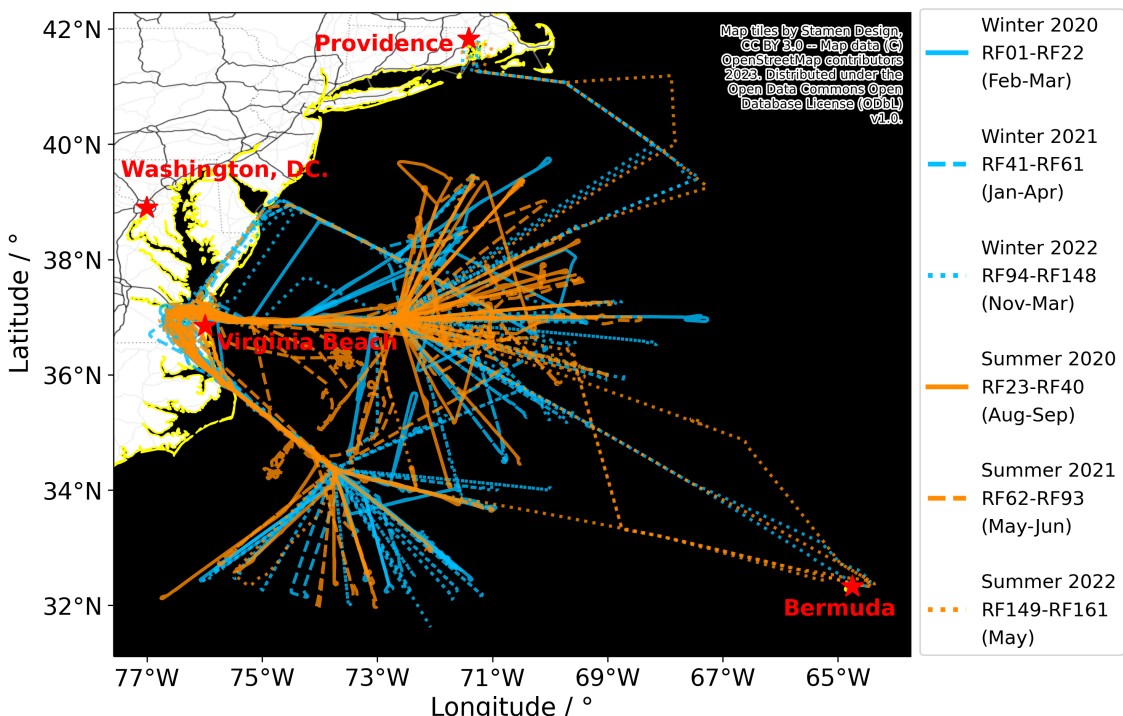

**Figure 1.** Flight tracks of 155 compiled research flights from the HU-25 Falcon during the 2020-2022 ACTIVATE mission. The number range of the research flights (RF) are given in the legend.

and ice ($ED_{ice}$) particles, liquid water content (LWC) and ice water content (IWC), with a focus on cloud phase. We relate the cloud properties to different meteorological conditions and derive the correlation of selected cloud properties in phase space diagrams and infer details about cloud formation, growth, and precipitation processes.

## 2 Instrumentation

The ACTIVATE campaign targeted the WNAO between 25 to 50°N and 60 to 85°W. Clouds, aerosols, trace gas, and meteorological data were measured simultaneously by two aircraft, with comprehensive dataset details provided elsewhere (Sorooshian et al., 2023). The HU-25 Falcon aircraft probed the MBL and lower troposphere with a pre-determined, repeated series of flight levels in a stairstepping fashion at different levels below cloud base, above cloud base, below cloud top, and above cloud top to provide a statistical approach to sampling the region (Sorooshian et al., 2019; Dadashazar et al., 2022). The King Air was instrumented with remote sensing instruments and dropsondes and operated above the HU-25 Falcon at around 9 km altitude. In total we compiled observations of 155 research flights with more than 512 flight hours from six ACTIVATE deployments. 17 Flights from June 2022 were omitted from this analysis that were based in Bermuda to maintain consistency in the sampled region, forthcoming work will explore the Bermuda flight data. In addition, two flights (RF38 and RF41) were omitted due

to partly missing cloud microphysical data. The flights cover three winter seasons (February-March 2020, January-April 2021 and November 2021-March 2022) and three summer periods (August-September 2020, May-June 2021 and May 2022), with flight tracks shown in Figure 1.

 ## 2.1 Cloud instruments

We use data from a cloud combination probe consisting of the Fast Cloud Droplet Probe (FCDP) and the Two-Dimensional Stereo (2D-S) probe, both developed by Stratton Park Engineering Company Incorporated (SPEC Inc.) (Lawson et al., 2019), onboard the HU-25 Falcon. The FCDP is a forward scattering single particle counter that measures sizes ranging between $1.5 - 50~\mu m$ (O'Connor et al., 2008; Knop et al., 2021). The FCDP measures the scattered light of particles passing a laser beam with a wavelength of 785 nm in a $4° - 12°$ scattering angle. The measured scattered light intensity is related to the particle diameter via Mie theory. Calibrations, with lime (n=1.52) and borosilicate (n=1.56) beads of known sizes, before and after each deployment verify the relationship between measured light intensity and particle diameter and validate the sizing. The FCDP is equipped with a pinhole-masked sizing detector for coincidence reduction (Lance, 2012) and special arm tips to reduce shattering (Korolev et al., 2013). In addition, methods for coincidence and shattering correction have been applied during post processing (Baumgardner et al., 1985; Field et al., 2006; Lawson, 2011; SPEC inc, 2012; Kleine et al., 2018). Particle number concentration is computed by multiplying the corrected count rate by the sample volume, which is the product of (i) a calibrated sample area of 0.294 $mm^2$ (at depth of field criterion 0.6) determined with a droplet generator experiment (Faber et al., 2018) and (ii) the probe air speed (PAS) measured by the Cloud Aerosol and Precipitation Spectrometer (CAPS, Voigt et al. (2021, 2022); Hahn et al. (2022); Moser et al. (2023)) flow tube at the opposite wing pot to relate the compressed measurement condition to ambient conditions (Weigel et al., 2016). The propagated uncertainties for scattering probes are $10 - 50\%$ in size and $10 - 30\%$ in cloud particle concentration ($N_C$) according to Baumgardner et al. (2017). Other studies found a size dependent uncertainty in sizing never exceeding 15% for the predecessor model Cloud Droplet Probe (CDP) (Lance et al., 2010; Faber et al., 2018; Braga et al., 2017a; Taylor et al., 2019). The FCDP has fast electronics with single particle counting similar to the Fast Forward Scattering Spectrometer Probe F-FSSP (Bräuer et al., 2021b, a). The fast electronics, per particle storage, and pinhole feature, result in FCDP in-cloud calibrated uncertainties in $N_C$/LWC/ED of 15%/40%/45% respectively.

The 2D-S is an optical array probe that generates shadow images of cloud particles on 128 photodiodes (Lawson et al., 2006, 2019). The measurement concept of optical array probes was developed by Knollenberg (1970) and measures diffraction patterns of particles passing the laser beam (Korolev et al., 1991). The 2D-S has a calibrated effective pixel resolution of 11.4 $\mu m$ and covers a diameter size range of $5.7 - 1465~\mu m$ (Lawson et al., 2019; Bansmer et al., 2018). The images contain shape information and can therefore distinguish between spherical and non-spherical particles. The diffraction pattern of particles changes with distance to the optical plane, which can be corrected in the spherical case (Korolev, 2007a). Non-spherical particles remain uncorrected with a systematic overestimation in size (Gurganus and Lawson, 2018). The ice mass of each particle is determined with the area-to-mass parametrization of Baker and Lawson (2006). The particle concentration is computed by multiplying the PAS with the photodiode array width which is the photodiode number times the effective pixel resolution and the size dependent depth of field at 50% intensity level (Korolev et al., 1998). According to Baumgardner et al.

(2017), the sizing and counting accuracy lies in a range of $10 - 100\%$ for optical array probes and the 2D-S, with the applied corrections and a relative fast response time of 41 ns, is estimated to be on the lower end in sizing spherical particles and in the middle for ice particles. The calibrated uncertainties in $N_C$/LWC/ED are 20%/50%/60% for the 2D-S in our case.

## 2.2 Cloud probe data evaluation

The FCDP and 2D-S combination probe measures particle size distributions with diameter between 3 and $1465$ $\mu$m and has an overlap in the size range of 5.7 to 50 $\mu$m. Since the counting efficiency of one or two pixel images in the 2D-S is smaller than particle images with more pixels (Korolev et al., 1998), we use the 2D-S starting with the third size bin of 28.5 to 39.9 $\mu$m. The FCDP has a significantly smaller sample volume compared to the 2D-S which is beneficial in terms of suppressing coincidence, but leads to an undercounting of larger particles due to their lower statistical occurrence in clouds observed during ACTIVATE

which cannot be resolved in a 1Hz sampling rate by the FCDP. Therefore we perform an overlap calculation which is outlined in the following. We use the nearest FCDP size bin of 27 to 30 $\mu$m next to the lower bin limit of 28.5 $\mu$m of the third 2D-S size bin and attribute all particles larger than 30 $\mu$m solely to the 2D-S and all particles smaller than 28.5 $\mu$m solely to the FCDP. The size distribution inside the third 2D-S size bin is unknown and therefore estimated with the next 2D-S size bin by linear interpolation. The interpolated distribution is used to split the third 2D-S size bin's number concentration into a portion of 28.5

to 30 $\mu$m attributable to the nearest FCDP size bin and a portion of 30 to 39.9 $\mu$m attributable to a new third 2D-S size bin with a reduced bin width of 9.9 $\mu$m. Then the number concentration of the nearest FCDP size bin is calculated with the arithmetic mean of the original and attributable number concentration if the original nearest FCDP size bin yields measured particles. If there were no measured particles in the nearest FCDP size bin the new nearest FCDP size bin is determined by the attributable number concentration of the third 2D-S size bin, since we expect the lack of particles is caused by the undercounting of larger

particles.

We derive microphysical cloud properties from the particle size distribution measured with the FCDP-2D-S cloud combination probe. In order to assess the particle phase, we assume all small particles <100 $\mu$m detected by the FCDP and the 2D-S to be liquid, as there is no other information available. This threshold has to be taken into account for model evaluation. For particles larger than 100 $\mu$m we use the phase information from the particle images to separate spherical and non-spherical

shaped particles. The lower detection limit for ice is 100 $\mu$m in diameter, because we use a minimum number of 50 pixels for habit classification. Korolev and Sussman (2000) showed that the minimum pixel number should be between 20 and 60 for the separation of irregulars and spheres. An adequate number of pixels is necessary to extract shape information with sufficient accuracy. We use the particle area of the 2D images of each ice particle to derive the ice water content using the method of Baker and Lawson (2006) and the maximum diameter for sizing. From the spherical particle size distribution, we derive the

number concentration ($N_{liquid}$), effective diameter ($ED_{liquid}$) and liquid water content (LWC) at 1 Hz resolution. The effective diameter is calculated as:

$$\mathrm{ED} = \frac{\sum\limits_{i} \mathrm{D}_i^3 \mathrm{N}_i}{\sum\limits_{i} \mathrm{D}_i^2 \mathrm{N}_i}, \tag{1}$$

where $\mathrm{D}_i$ is the arithmetic mean diameter and $\mathrm{N}_i$ the number concentration of the respective size bin $i$ (Parol et al., 1991). The LWC is calculated from the particle size distribution based on number and particle size of the droplets:

$$\mathrm{LWC} = \sum_{i} \mathrm{LWC}_i = \sum_{i} \frac{\mathrm{N}_i}{\mathrm{V}} \frac{4}{3} \pi \rho_{\mathrm{w}} \left( \frac{\mathrm{D}_i}{2} \right)^3, \tag{2}$$

where $\rho_{\mathrm{w}}$ is the density of water and the water droplets are assumed to be an ideal sphere. V is the probed volume. $\mathrm{LWC}_i$ represents the liquid water content of the respective size bin $i$. Similarly, we derive the ice number concentration ($\mathrm{N}_{\mathrm{ice}}$), effective diameter ($\mathrm{ED}_{\mathrm{ice}}$) and ice water content (IWC) from the non-spherical particle size distribution measured by the 2D-S for particles $>100$ $\mu$m. The IWC is calculated as:

$$\mathrm{IWC} = \frac{\sum\limits_{p} \mathrm{M}_{\mathrm{Eis},p} w_p}{\mathrm{V}} = \frac{\sum\limits_{p} 0.115 \mathrm{A}_p^{1.218} w_p}{\mathrm{V}}, \tag{3}$$

whereby all particles $p$ in the measurement volume are summed up. $\mathrm{M}_{\mathrm{Eis},p}$ is the mass-dimension relationship of Baker and Lawson (2006) and $\mathrm{A}_p$ the area of the particles. The factor $w_p$ represents a weighting of the chosen method. Here, we apply the *All-In* method of Knollenberg (1970). The uncertainties for the FCDP-2D-S cloud combination probe in $\mathrm{N_C}$/LWC/ED are 15%/40%/45% in cloud and are consistent with the FCDP uncertainties, since in over 95% of the in cloud measurements more than 98% of all cloud particles are measured by the FCDP. The corresponding uncertainties for precipitation in $\mathrm{N_C}$/LWC/ED are 55%/85%/85% and are driven by low statistics where the statistical uncertainty is estimated with Poisson statistics (Baumgardner et al., 2017).

### 2.2.1 Phase space diagrams of WNAO Boundary Layer Clouds

To investigate microphysical properties of clouds we have to define a cloud in the context of the in-situ data. Phase space diagrams of selected cloud properties are useful for identifying precipitation, determining cloud thresholds, and facilitating the investigation of in-cloud and out-of-cloud samples. Finally, we suggest to use a phase space diagram to compare in-situ measured cloud properties to results from process and global models. Figure 2a depicts the distribution of $\mathrm{ED}_{\mathrm{liquid}}$ versus that for $\mathrm{N}_{\mathrm{liquid}}$ for all winter deployments based on 1 Hz particle data from the FCDP-2D-S combination probe; see Figure S1 for all summer deployments.

The region with $0.03 < \mathrm{N}_{\mathrm{liquid}} < 10$ cm$^{-3}$ and $\mathrm{ED}_{\mathrm{liquid}} < 20$ $\mu$m can be attributed to background aerosols (gray) in the MBL out of clouds with low number concentrations of rather large volatile or non-volatile aerosol. The lower limit of $\mathrm{N}_{\mathrm{liquid}}$ reflects the FCDP detection limit in number concentration. Liquid clouds (orange) are associated with higher $\mathrm{N}_{\mathrm{liquid}}$

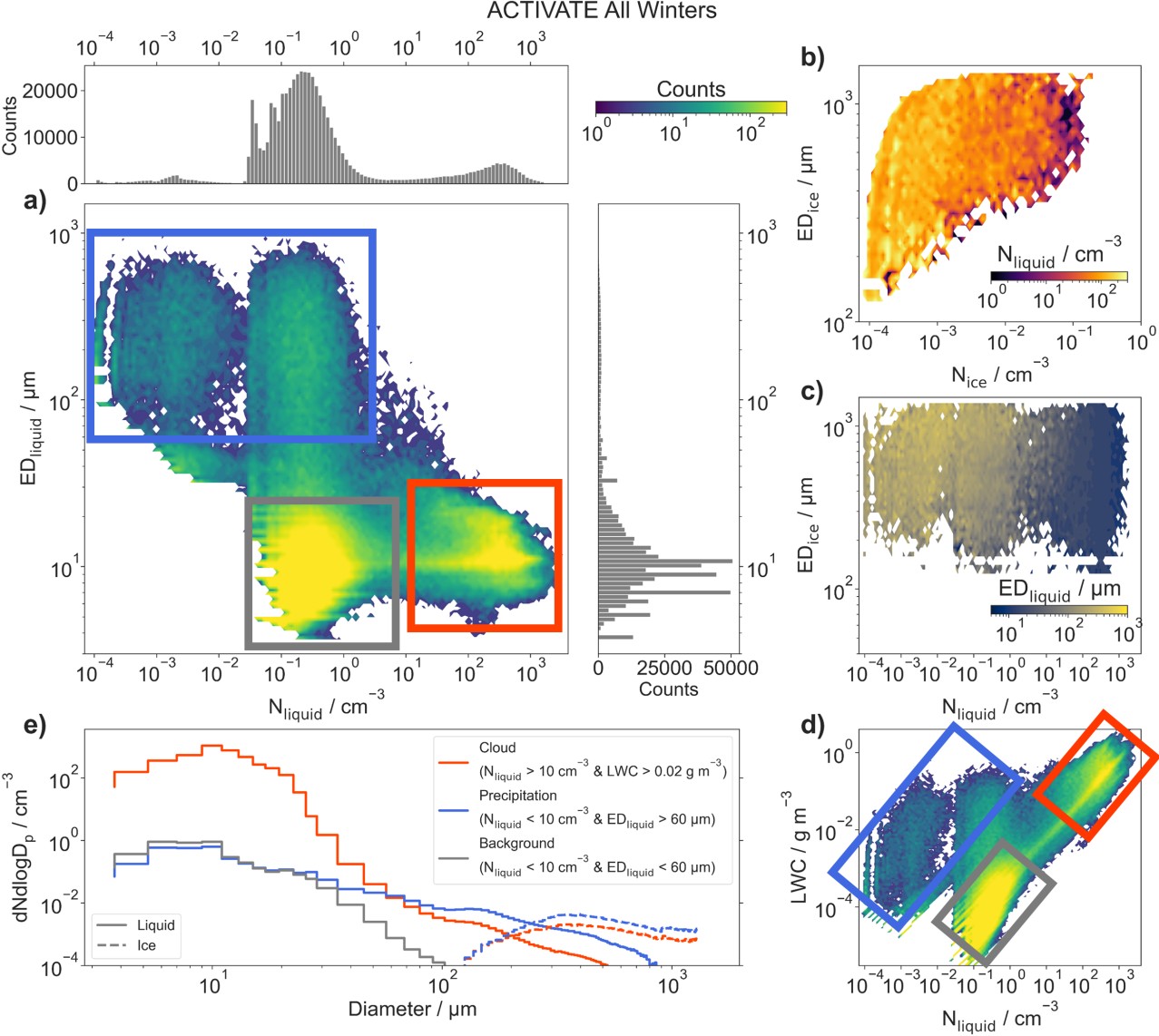

**Figure 2.** Cloud (orange), precipitation (blue), and background (gray) illustrated by the occurrence frequency of cloud properties in parameter phase space of $ED_{liquid}$ and $N_{liquid}$ (a) and parameter phase space of $ED_{ice}$ and $N_{ice}$ (b), $ED_{ice}$ and $N_{liquid}$ (c) and LWC and $N_{liquid}$ (d). The color code shows the number of seconds of cloud data and is the same for panels (a) and (d). The color code of panel (b)/(c) shows the mean $N_{liquid}/ED_{liquid}$ values. Mean particle size distribution of the cloud, precipitation, and background region (e). All subplots relate to all winter deployments.

in the range of 100 to 1000 $cm^{-3}$ and $ED_{liquid} < 30~\mu m$; those clouds were frequently encountered. Data with $N_{liquid}$ below 3 $cm^{-3}$ and $ED_{liquid}$ above 60 $\mu m$ up to one millimeter indicate the detection of precipitation (blue). This suggests that $ED_{liquid}$

is well suited to distinguish precipitation from cloud liquid droplets and $N_{liquid}$ helps to distinguish between inside or outside of clouds. Figure 2b shows the phase space diagram for ice particles identified by the 2D-S with its lower size detection limit of $100\,\mu$m and represents precipitation of ice particles or snow. The color code shows that high $ED_{ice}$ and $N_{ice}$ values correlate with lower $ED_{liquid}$ values. Lowest $N_{ice/liquid}$ values indicate the detection limit of the 2D-S which is significantly smaller due to a larger sample volume of the probe. $ED_{ice}$ is filled only when ice particles are detected by the 2D-S, and is thus a good indicator for the presence of ice. Its relation to $N_{liquid}$ instead of $N_{ice}$ (Figure 2c) suggests that ice exists both inside as well as outside of clouds.

In Figure 2d the phase space of LWC and $N_{liquid}$ exhibits the same regions for liquid cloud, precipitation, and background. In contrast, a fixed LWC threshold is often used in literature for defining clouds (McFarquhar et al., 2007; Ahn et al., 2018; Abel et al., 2020; Korolev and Milbrandt, 2022). High LWC values could either stem from large particle concentrations of small particles typical for liquid clouds or low concentrations of large particles linked to precipitation. This suggests that LWC only is not a good indicator for separating cloud from precipitation without $ED_{liquid}$, because a typical LWC threshold of $0.02\,\mathrm{gm}^{-3}$ in the ACTIVATE campaign includes not only the cloud data, but also a fraction of precipitation. Even more important is the fact that a lot of data are missed if precipitation is targeted only by the LWC threshold. Additionally, a lower fixed LWC threshold for the ACTIVATE campaign hampers the differentiation of aerosol from precipitation events.

Given the disadvantages of using a single LWC-based threshold, we add a threshold for $N_{liquid}$ similar to previous studies (e.g. Wood, 2005; Gupta et al., 2021; Dzambo et al., 2021), because it provides a better differentiation between in-cloud and out-of-cloud situations and avoids a misclassification of precipitation. Here we use the cloud threshold $N_{liquid} > 10\,\mathrm{cm}^{-3}$ and LWC $> 0.02\,\mathrm{gm}^{-3}$. The remaining data are categorized with the $ED_{liquid}$ either above or below $60\,\mu$m into the precipitation or background group, respectively. The groups' mean particle size distribution of liquid droplets and ice particles are illustrated in Figure 2e. We want to note that the use of the means instead of medians emphasizes outliers and that a relative occurrence of 10% in measurement seconds is equivalent to a shift of one magnitude towards lower size bin concentrations. The cloud group with $ED_{liquid}$ around $10\,\mu$m includes occasional measurements of larger particles showing the development of precipitation inside cloud. Finally, we define cloud events as periods where the HU-25 Falcon flight has the prescribed cloud threshold flag set for > 1 consecutive seconds. We calculate mean microphysical cloud properties for each cloud event. In mixed-phase clouds, we give both the liquid and the ice information as percentage of seconds containing ice in relation to the duration of the cloud event.

## 3 Results and Discussion

### 3.1 Occurrence and Microphysical Properties of Marine Boundary Layer Clouds in the WNAO

We first show an overview of the microphysical cloud properties of all marine boundary layer cloud events measured in the WNAO during the ACTIVATE campaign in Figure 3. After this general description, we will examine cloud properties and processes in higher detail in the subsequent sections. Figure 3 shows the LWC of each cloud event versus altitude for the six deployments. Clouds were observed at altitudes between $0.3$ and $4$ km in and above the boundary layer.

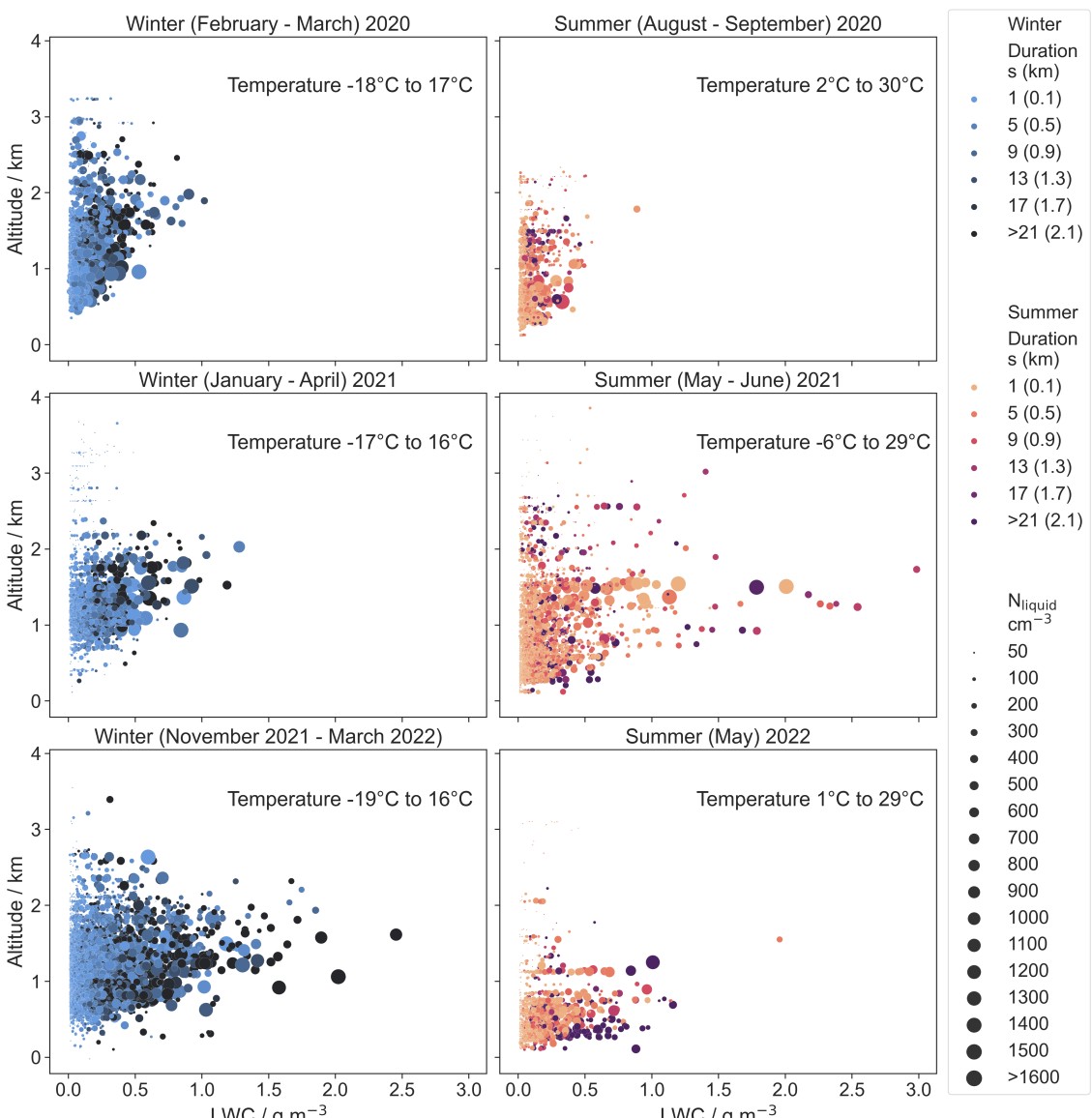

**Figure 3.** Altitude dependence of the liquid water content from all cloud events during the ACTIVATE mission in winter and summer 2020, 2021 and 2022. Each dot represents a cloud event with their duration and corresponding width in parenthesis color coded in winter from light blue to dark blue and in summer from orange to violet. The width of the cloud event is calculated from its duration with a representative HU-25 Falcon true air speed of $100 \text{ m s}^{-1}$. The mean number concentration of each event is given with the dot size.

The LWC of the WNAO boundary layer clouds increase with altitude is in line with the assumption of idealized cloud formation and growth due to adiabatic uplift. In an upward motion, the air cools down and cloud condensation nuclei are

activated into water droplets as soon as the Kelvin barrier is surpassed and water supersaturation is reached (Feingold and

**Table 1.** Cloud event statistics for each deployment during the ACTIVATE mission. Average number of cloud events per RF ($\varnothing$) during a deployment. Width, LWC, $N_{liquid}$ and $ED_{liquid}$ are given in median/mean and their standard deviation in parenthesis.

| Deployment | Total Events | $\varnothing$ | Width km | LWC g m$^{-3}$ | $N_{liquid}$ cm$^{-3}$ | $ED_{liquid}$ $\mu$m |
|---|---|---|---|---|---|---|
| Winter 2020 | 2690 | 122 | 0.3/0.7($\pm$1.4) | 0.06/0.10($\pm$0.11) | 186/249($\pm$223) | 8.8/11.7(11.7) |
| Winter 2021 | 2126 | 133 | 0.3/0.7($\pm$1.0) | 0.11/0.15($\pm$0.14) | 152/227($\pm$253) | 11.3/14.1(10.9) |
| Winter 2022 | 6683 | 122 | 0.3/0.8($\pm$1.8) | 0.12/0.19($\pm$0.21) | 113/180($\pm$193) | 11.9/14.8(8.9) |
|  | 6281[*] | 119[*] | 0.3/0.8($\pm$1.8)[*] | 0.12/0.19($\pm$0.21)[*] | 113/178($\pm$193)[*] | 11.9/14.7(8.7) |
| Summer 2020 | 1517 | 89 | 0.2/0.4($\pm$0.6) | 0.05/0.08($\pm$0.08) | 114/163($\pm$165) | 9.8/11.8(8.0) |
| Summer 2021 | 3127 | 98 | 0.2/0.6($\pm$2.2) | 0.09/0.16($\pm$0.22) | 84/118($\pm$140) | 13.2/16.5(16.1) |
| Summer 2022 | 1253 | 96 | 0.2/1.4($\pm$5.4) | 0.10/0.16($\pm$0.19) | 93/132($\pm$130) | 12.8/14.5(6.4) |
|  | 835[†] | 84[†] | 0.2/0.6($\pm$2.3)[†] | 0.08/0.14($\pm$0.16)[†] | 94/129($\pm$129)[†] | 12.7/13.2(2.2) |

[*] Without RF128/RF129 (stratiform cloud deck).

[†] Without RF150/RF151/RF152 (stratiform cloud deck).

Heymsfield, 1992; Ervens et al., 2005). With further updraft, more water vapor condenses and the droplets grow, which leads to an increase in LWC, as indicated by the increasing maximum LWC values with altitude in Figure 3. Due to the sampling strategy in the MBL during ACTIVATE, we lack the exact cloud base height for all cloud events and therefore refrain from the calculation of adiabatic LWC. However, the linear increase of LWC with height is consistent with adiabatic expectations.

A total of 11499 cloud events were measured in winter and 5897 in summer, with the detailed statistics of the cloud events in Table 1 and the full distribution of LWC and $N_{liquid}$ for each deployment in Figure S2. We compute cloud width from the mean duration of a cloud event, and an in-situ derived horizontal flight cloud cover from both the cloud width and $\varnothing$. We want to note that the duration and occurrence of the cloud events depend on the flight patterns. The clouds were probed by the same flight pattern in the majority of cases and the flight patterns rarely changed due to unforeseen reasons. Therefore the statistics from the 155 flights is sufficient for a trend analysis. The less frequent cloud observations per research flight in summer in combination with the similar (summer 2021) and reduced (summer 2020 and summer 2022[†]) horizontal cloud widths compared to winter indicate smaller clouds and enhanced broken cloud systems in summer. Consequently, the larger distances between clouds is equivalent to more cloud free areas, which results in a reduced flight cloud cover during the summer. In contrast, the large cloud width in summer 2022 stems from stratiform closed cloud decks in stratus during RF150/RF151 (5 May 2022) to Providence, Rhode Island, and during RF152 (10 May 2022). The aerosol background in the WNAO above 39°N deviates from areas farther south with higher aerosol abundance (Corral et al., 2021), which could result in more frequent occurrence of stratiform cloud decks.

     The timeframe varies between the the summer seasons. LWC, $N_{liquid}$ and $ED_{liquid}$ of summer 2020 suggests a particle size distribution with a dominant small particle mode below 50 $\mu$m and less large particles in late summer. In contrast, observations during early summer 2021 and 2022 feature a more pronounced large particle mode, which probably stems from enhanced

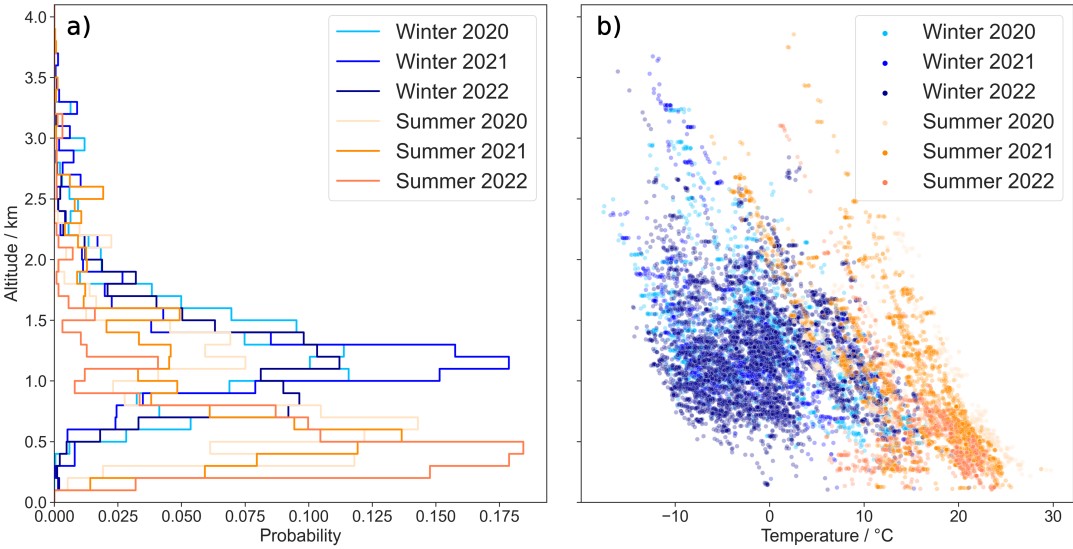

**Figure 4.** Probability distribution of the cloud occurrences versus altitude (a) and the corresponding temperature distribution (b) for all cloud events in winter and summer 2020, 2021 and 2022.

coalescence processes promoting warm rain formation (Berry and Reinhardt, 1974; Pinsky et al., 2000; Rauber et al., 2007; Braga et al., 2017b). Kirschler et al. (2022) found that smaller $N_{liquid}$ and LWC values in summer 2020 relative to winter 2020 can be explained by increased updraft speeds in winter and consequently increased supersaturation at the cloud base, despite higher aerosol number concentrations with sizes >85 nm than in summer. The positive correlation of updraft speed and

255 supersaturation at the cloud base determines the critical diameter of aerosol activation leading to a higher fraction of activated aerosols at smaller critical diameters for higher updraft speeds (Köhler, 1936; Dusek et al., 2006; Schulze et al., 2020; Braga et al., 2021a).

The probability of cloud occurrence with height is shown in Figure 4a. Performing flight levels at different altitudes limits the number of times we penetrate the cloud base, but allows us to show the cloud occurrence versus altitude. In addition,

the typical cloud field consisted of broken cumulus clouds with varying cloud bases. Clouds are typically confined to the MBL, with a peak of occurrence of around 1.3 km in winter and 0.5 km in summer. This difference can be explained by the MBL temperature inversion height, which caps the vertical propagation of the clouds and determines the maximum cloud top height. The higher MBL temperature inversion height is connected to the larger surface heat fluxes in winter, resulting from the temperature difference between cold and dry air masses from the west and the warm Gulf Stream, which increases updrafts

and turbulence, thereby deepening the marine boundary layer (Small et al., 2008; Chelton et al., 2004). While the majority of the MBL clouds are at 0.5 km in summer, there is a second layer at 1.3 km height which is most pronounced in summer 2020 suggesting a double (decoupled) cloud layer structure. The increased occurrence of clouds at higher altitudes in winter

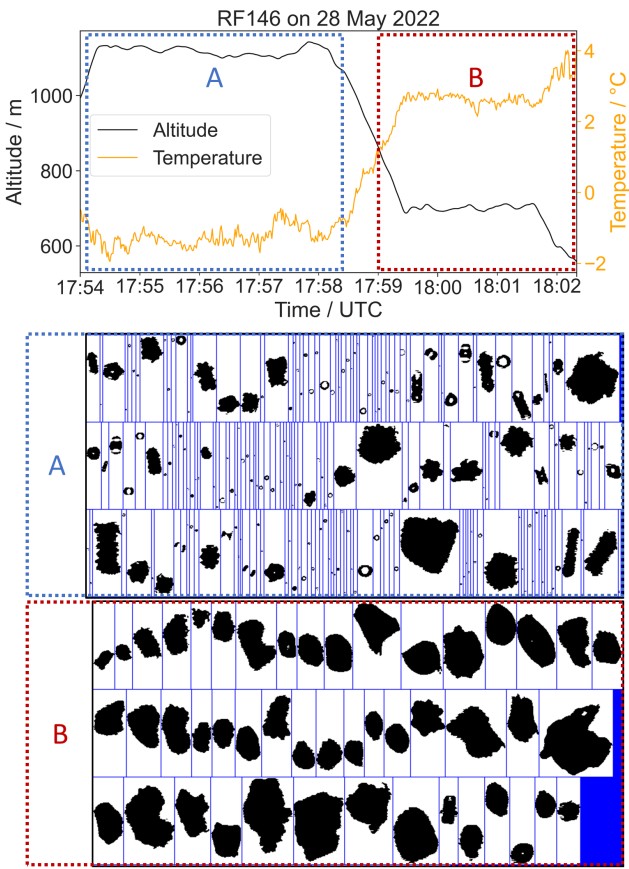

**Figure 5.** 2D-S Graupel measurements above 0°C during RF146 on 28 May 2022. Mixed-phase conditions above cloud base in Box A (blue) and graupel measurements in the following near cloud base leg at temperatures up to 3.5°C in Box B (red).

than in summer is consistent with the satellite climatology in Painemal et al. (2021). In addition, the temperature distribution of the clouds is shown in Figure 4b. In winter, the clouds were observed at higher altitudes and colder temperatures compared

to summer, which mainly exhibited temperatures above 0°C. During summer 2021, mixed-phase clouds were only measured in the morning research flight on 14 May 2021 (RF63). Other research flights during summer 2021 and 2022 contained non-spherical particles at temperatures above 10°C which are mainly larger than 100 $\mu$m. This might indicate the presence of large bioaerosol particles in small concentrations (see examples in Figure S3). The winter and summer 2021 cloud events above 2 km with temperatures below 0°C contained ice particles and were therefore classified as mixed-phase or ice clouds. In all

winter seasons ice particles, mostly graupel, were observed. A case of graupel measurements above 0°C up to 3.5°C is shown in Figure 5. They probably formed near the colder cloud top before falling into the warmer cloud base without enough time to melt completely.

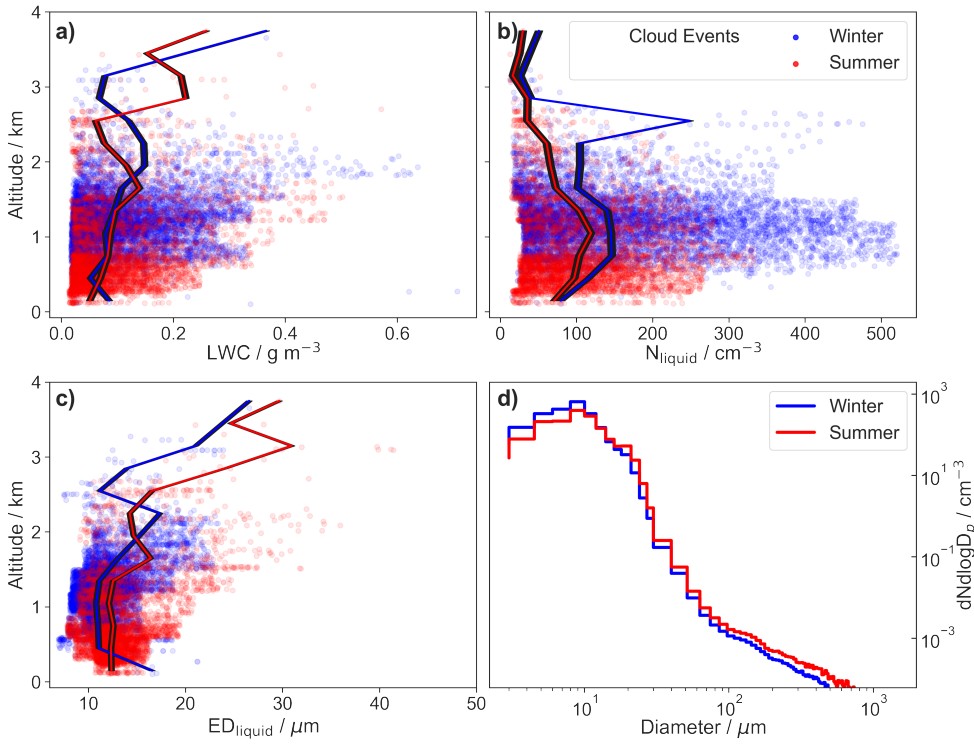

**Figure 6.** Median vertical profiles (lines) of LWC (a), $N_{liquid}$ (b) and $ED_{liquid}$ (c), and the mean particle size distribution (d) of all pure liquid cloud events in all winter (blue) and summer (red) seasons. Cloud events (dots) in the $10 - 90\%$ percentile range of a 300 m altitude intervals starting at the surface are shown.

## 3.2 Formation and Properties of Liquid Marine Boundary Layer Clouds in the WNAO

We now investigate the properties of pure liquid clouds in more detail. To this end we show vertical profiles of LWC, $N_{liquid}$ and
$ED_{liquid}$ as well as average particle size distributions of liquid clouds in Figure 6. The majority of cloud events were measured below 1.8 km in winter and below 0.8 km in summer. Statistical inferences above 2 km are limited due to the reduced number of cloudy samples. The LWC increase with altitude up to 1.6 km is comparable in both seasons and differs above. $N_{liquid}$ is higher in winter, which is the results of higher updraft speeds in winter, leading to stronger activation of cloud condensation nuclei (Kirschler et al., 2022). In winter and summer, $N_{liquid}$ increases to a maximum near 1 km altitude. Above that, $N_{liquid}$
decreases in both seasons except for a cloud event at 2.7 km in winter. This is in line with an increase in $ED_{liquid}$ and LWC at altitudes above 1 km. The steady increase of $ED_{liquid}$ with altitude in Figure 6 shows the growth processes by uptake of water from the gas phase and by collision-coalescence of liquid particles inside the clouds for all seasons. The summer periods have larger $ED_{liquid}$ and corresponding lower $N_{liquid}$. The higher $N_{liquid}$ and smaller $ED_{liquid}$ of liquid marine boundary layer clouds increase cloud lifetime and suppress processes leading to precipitation and thus yielding lower concentrations for
droplets larger than 40 $\mu$m during the winter as compared to summer (Albrecht, 1989; Freud et al., 2011; Freud and Rosenfeld,

2012; Braga et al., 2021b). Above 2.3 km, midlevel clouds were observed, but their lower measurement frequency hampers a more detailed trend analysis.

## 3.3 Formation and Properties of Mixed-Phase Marine Boundary Layer Clouds in the WNAO

Figure 7 shows the vertical profiles of mixed-phase cloud events. Since the summer 2021 deployment only contained one research flight with mixed-phase clouds above 2 km, we will focus our study solely on the winter seasons and compare the liquid fraction of the cloud to their equivalent parameter in pure liquid clouds. We would like to remind the reader that, based on the 2D-S shape information, we assume particles with sizes smaller than 100 $\mu$m to be liquid, and liquid or ice for larger sizes. This could lead to an overestimation of liquid fraction and underestimation of ice fraction in clouds. The mixed-phase clouds could therefore contain a fraction of pure ice clouds. $N_{liquid}$ in mixed-phase clouds shows a similar pattern to that for liquid clouds with an increase up to about 1.3 km and a subsequent decrease. The $ED_{liquid}$ has a wider range from 5 to 80 $\mu$m than in the case of pure liquid clouds and could contain an ice fraction, which is not distinguishable with the current instrumentation. $N_{ice}$, IWC, and $ED_{ice}$ all increase above 1 km up to 3.5 km in the mixed-phase and ice clouds in and above the MBL. Due to the limited statistics, the ice parameters show a strong fluctuation in their mean values with altitude. In winter, ice particles are measured above 2 km almost during the entire mixed-phase cloud segments. The particle size distribution of the ice particles measured by the 2D-S shows the abundance of ice particles up to 1.5 mm, which is the upper detection limit of the 2D-S.

The median vertical $ED_{ice}$ profile features an increase with altitudes which can be attributed to the WBF mechanism primarily observed in mixed-phase clouds (Wegener, 1911; Bergeron, 1935; Findeisen, 1938; Korolev, 2007b). The ice fraction is defined as the percentage of time containing ice particles relative to the duration of the individual cloud event and shows increasing glaciation of the clouds with altitude above 1.5 km and correlates with the onset of the WBF process. In addition, the LWC and $N_{liquid}$ decrease as the WBF process converts liquid into ice mass. Mixing processes with dry air from the free troposphere affect the $N_{liquid}$ and LWC first and small droplets evaporate faster due to their size and the Kelvin effect, while ice crystals survive longer. Increased mixing owing to free tropospheric air entrainment was observed in the WNAO farther off shore (Tornow et al., 2022).

The mixed-phase cloud size distribution reveals that the WBF process and other mixed-phase related processes like coalescence, riming, and aggregation (Böhm, 1992b, a) result in higher concentrations of large particles >100 $\mu$m compared to the size distribution of pure liquid clouds. These processes lead to precipitation caused by the ice phase. In addition, the ice particles trigger other processes of secondary ice production like droplet fragmentation, ice fragmentation through thermal shock or during sublimation, and splintering during riming and ice collision, which accelerate the process even further (Hallett and Mossop, 1974; Field et al., 2017b; Keinert et al., 2020; Korolev et al., 2020, 2022). Furthermore, the MBL is turbulent (Brunke et al., 2022) and the cloud experiences spatially small-scale upwelling and downwelling that can maintain non-equilibrium mixed-phase cloud state for several hours (Korolev, 2008; Yang et al., 2015).

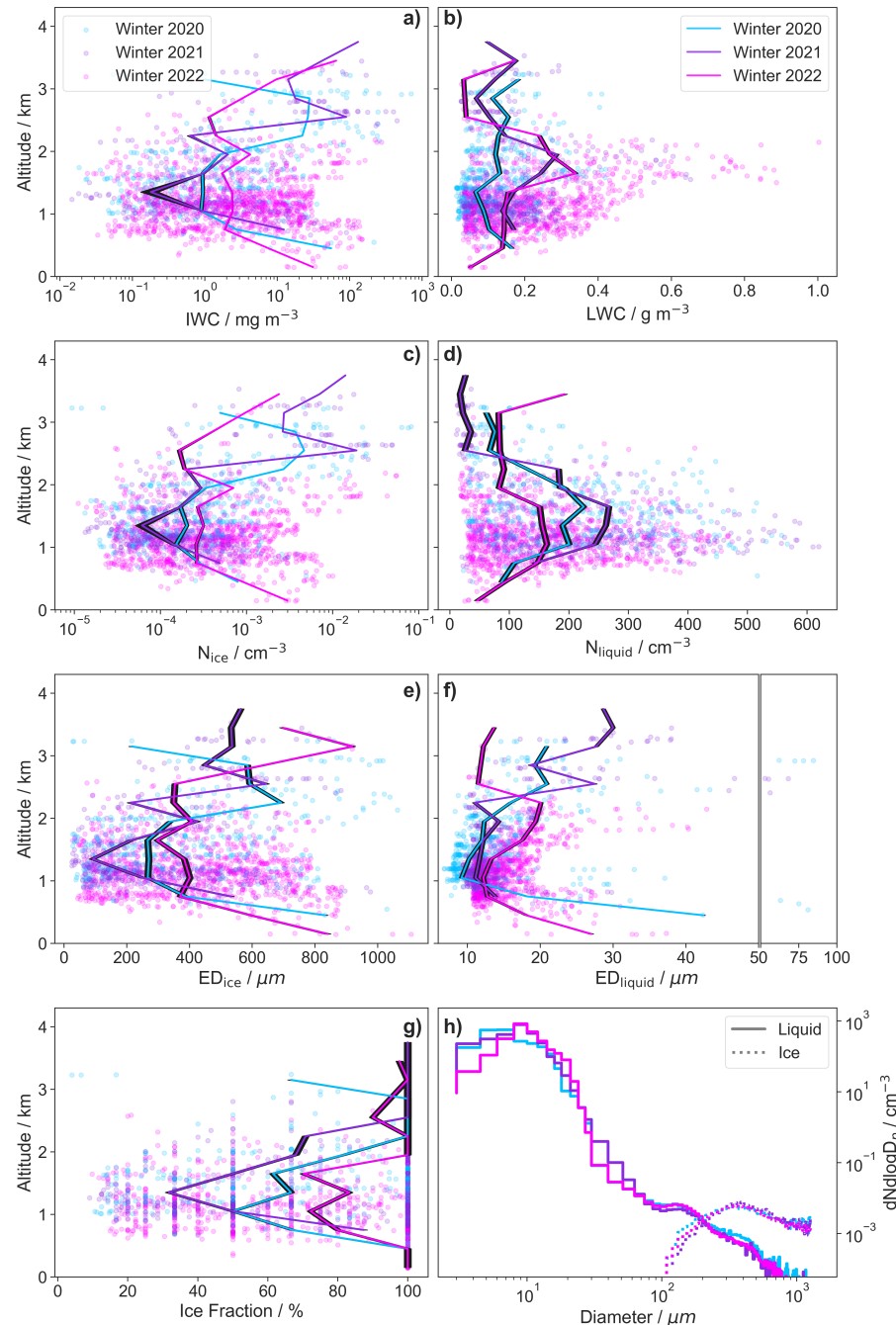

**Figure 7.** Median vertical profiles (lines) of LWC (b), IWC (a), $N_{liquid}$ (d), $N_{ice}$ (c), $ED_{liquid}$ (f) (vertical double line indicates a switch in the scaling of the x-axis), $ED_{ice}$ (e) and the ice fraction (g) for all mixed-phase cloud events in winter 2020, 2021 and 2022. Cloud events (dots) in the $10 - 90\%$ percentile range of $300$ m altitude intervals starting at the surface are shown. The mean particle size distribution (h) of retrieved liquid (lines) and ice (dotted) particles.

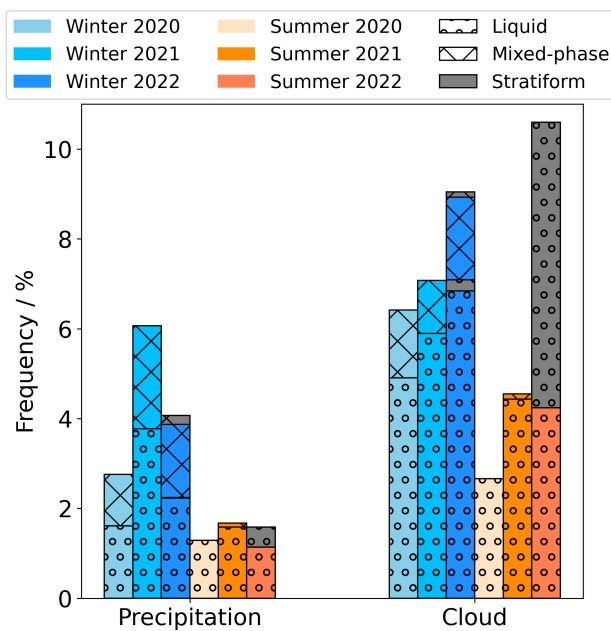

**Figure 8.** Frequency of clouds and precipitation across the entire flights for winter and summer 2020, 2021 and 2022. The cloud phase is indicated by circles (liquid) and crosses (mixed-phase). The contribution of flights with a stratiform cloud deck, see Table 1, is in gray color.

### 3.4 Statistics of Cloud and Precipitation Measurement Data

After the discussion of liquid and mixed-phase clouds, we now investigate precipitation measurements and provide an overview of their occurrence and spatial distribution. Figure 8 shows the frequency of precipitation and cloud measurements on a per second basis in winter and summer 2020, 2021, and 2022, using all seconds that fall into the precipitation ($N_{liquid} < 10$ cm$^{-3}$ and $ED_{liquid} > 60$ $\mu$m) or cloud ($N_{liquid} > 10$ cm$^{-3}$ and LWC $> 0.02$ g m$^3$) category. The frequency is defined as the ratio of precipitation or cloud measurements to all measurement seconds. A further distinction of the phase with 2D-S shape information shows the respective liquid and mixed-phase or ice fraction. In winter, the total occurrence of cloud measurements is between 6.5% and 9%, while in summer a much larger spread of 3% to 11% is observed. In the summer and winter 2022 deployments, flights north of 39°N with stratiform cloud decks contribute substantially to the observed cloud frequency, since measurement seconds are used and considerably more flights were conducted in winter (55 RF) than in the summer (13 RF) of 2022. The findings from the observation of defined cloud events are reflected in the frequency of cloud measurements with less frequent cloud measurements during summer compared to the winter seasons. The cloud deck is consequently characterized by more cloud-free areas and the width of the clouds is reduced. The difference between summer 2020 and summer 2021/2022 (without flights that contain a stratiform cloud deck) could stem from the different time periods sampled during each deployment (August-September 2020 and May-June 2021/2022). A similar reduction in cloud fraction from early to late summer was seen from a combined CloudSat and CALIPSO analysis (Painemal et al., 2021). We want to emphasize that in general the

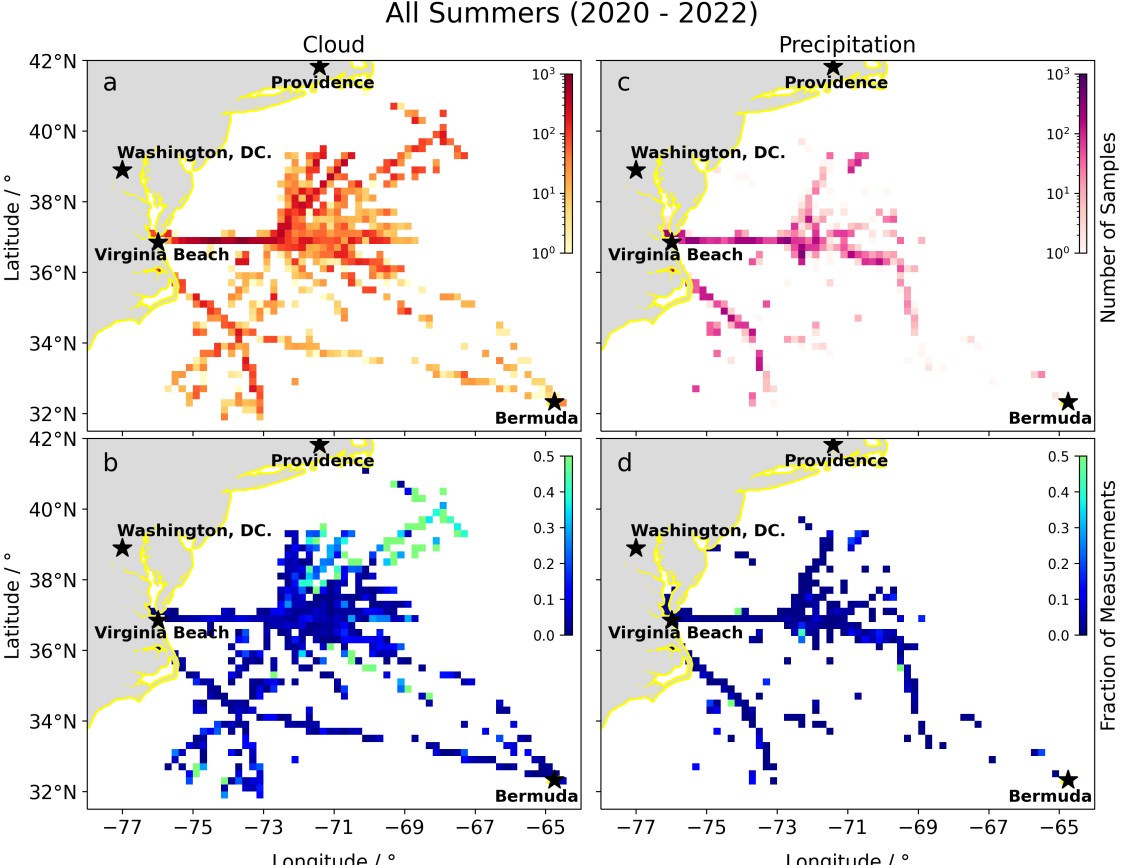

**Figure 9.** The spatial distribution in 0.2° latitude and longitude grid cells of cloud (left) and precipitation (right) measurements. The upper panels show the number of cloud measurement seconds (a) in oranges and the number of precipitation measurement seconds (c) in violets. The lower panels show the fraction of cloud (b) and precipitation (d) measurements color coded from dark blue to green. All subplots relate to all summer deployments.

in-situ derived flight cloud cover cannot be directly compared to satellite cloud cover. ACTIVATE is a special case because the same strategy was used for each flight, large statistics are available, and the width of the predominant cumulus clouds was

340 mainly within the cloud flight leg. The occurrence of mixed-phase clouds varies between 1% and 2% in winter.

The frequency of precipitation in Figure 8 shows larger values in winter (3 - 6%) than in summer (1.3 - 1.6%), in qualitative agreement with a satellite microwave climatology in Painemal et al. (2021). In addition, the relative mixed-phase fraction in precipitating samples is higher than that in non-precipitating clouds. This increased proportion of mixed-phase precipitation is in line with the observed larger number concentrations of particles above 100 $\mu$m in mixed-phase clouds which consequently

promotes precipitation. The spatial distribution of precipitation and cloud measurements in summer is shown in Figure 9. Most of the measurements in summer were made in the west-to-east corridor. Cloud measurements in summer are distributed all

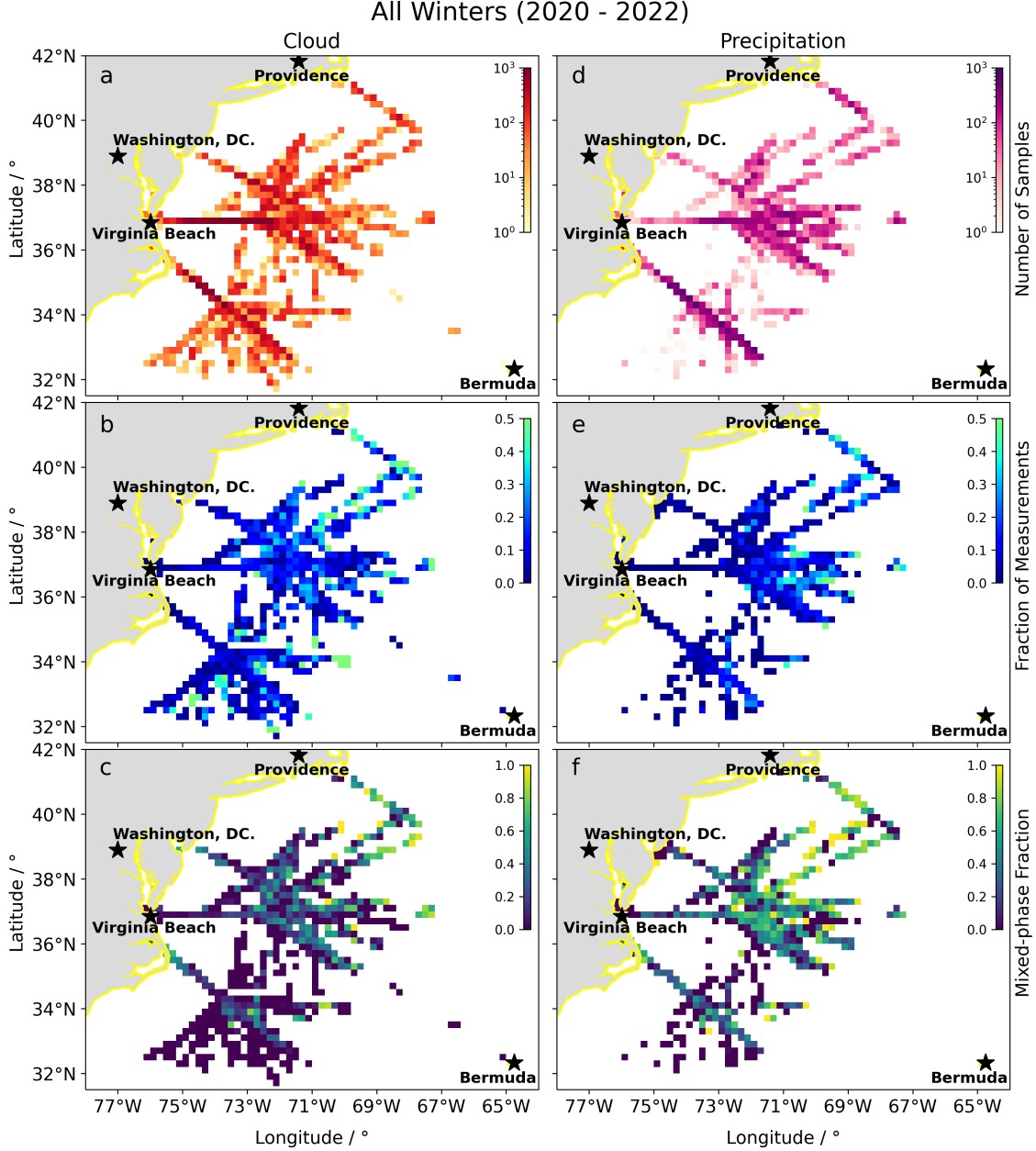

**Figure 10.** The spatial distribution in $0.2°$ latitude and longitude grid cells of cloud (left) and precipitation (right) measurements. The upper panels show the number of cloud measurement seconds (a) in oranges and the number of precipitation measurement seconds (d) in violets. The middle panels show the fraction of cloud (b) and precipitation (e) measurements color coded from dark blue to green. The lower panels show the fraction of mixed-phase clouds (c) and precipitation (f) with coloring from violet to yellow. All subplots relate to all winter deployments.

along the flight paths, whereas precipitation was very rarely measured between the west-to-east and northwest-to-southeast corridors and north of 37.5˚N. The low-level cloud fraction, computed over 0.2˚ regular grids, features values that exceed 40% north of 38.5˚N and east of 71˚W, whereas cloud fractions remain below 10% for the rest of the domain. Low cloud fractions near the coast occur over with a local minimum in sea surface temperature, west of the Gulf Stream (Painemal et al., 2021). Spatial distribution of precipitation and cloud measurements in winter are depicted in Figure 10. As in summer, most measurements were made along the west-to-east and northwest-to-southeast corridors. When the HU-25 Falcon followed the west-to-east corridor, then precipitation measurements are seen almost at all locations where cloud measurements were observed. In contrast, in the northwest-to-southeast corridor, precipitation measurements occur mostly in the southeast direction. The relative cloud frequency shows that clouds occur more frequently in winter than in summer by 20 - 30%, somewhat consistent with satellite climatologies (Dadashazar et al., 2021b; Painemal et al., 2021). The ACTIVATE flights were mostly conducted over the Gulf Stream, and the increased frequency of cloud measurements in winter correlates with high surface latent and sensible heat fluxes driven by the Gulf Stream, which peak in winter and are considerably lower in summer (Painemal et al., 2021). In terms of spatial distribution, cloud frequency tends to increase east of 74˚W, whereas precipitation measurements occur more often east of 71.5˚W. This eastward shift in precipitation occurrence is linked to the dynamics of cyclonic systems, with boundary layer clouds developing west of the cyclone (postfrontal) and precipitation being more prevalent over the eastern edge of the postfrontal region, where synoptic-scale ascending motions are more common (Painemal et al., 2023). An increased eastward occurrence of precipitation was observed in typical WNAO cloud conditions and in CAOs (Dadashazar et al., 2021a; Tornow et al., 2021). The importance of winter midlatitude weather disturbances in cloud and precipitation occurrence can be assessed by applying the synoptic classification of Painemal et al. (2023) to the individual research flights. 52% of the winter flights are characterized by a mid-tropospheric trough configuration, which either encompasses the entire WNAO region or is limited to the WNAO western section. These cases are typically associated with CAO conditions of varying intensity (Painemal et al., 2023). This synoptic pattern features lower-tropospheric subsidence and northerly/northwesterly winds that enhance surface heat turbulent fluxes (Painemal et al., 2023), and, thus, likely strengthening the boundary layer turbulence. All these conditions are favorable for the formation of low-level clouds and offshore precipitation near the western sector of midlatitude cyclones (Painemal et al., 2023). Moreover, updrafts driven by strong turbulence are thought to be a key factor that explains the significant enhancement of cloud droplet number concentration in winter relative to summer (Kirschler et al., 2022). In contrast, summer synoptic variability is less pronounced, and characterized by midlatitude geopotential height perturbations with magnitude 50% smaller than those observed in winter. This gives rises to a semipermanent anticyclone that undergoes synoptic changes in its magnitude and extension. The summer anticyclonic circulation drives weak surface heat turbulence fluxes and reduced static stability under a warm sea surface, promoting the occurrence of shallow cumulus clouds with low spatial cloud coverage (Painemal et al., 2021).

The mixed-phase fraction of clouds and precipitation measurements in Figure 10c,f show a latitudinal decline with highest fractions north of 37.5˚N. It is defined as measurement seconds containing ice particles to all cloud or precipitation measurements in the regular grids, respectively. The mixed-phase fraction of the precipitation measurements is higher than that of the cloud measurements. This may be due to the lower detection limit for ice classification of the 2D-S. It is likely that the

cloud measurements had a higher fraction of mixed-phase or ice clouds that could not be detected. In those cloud measurement seconds, no large ice particles were measured by the 2D-S. Thus, smaller ice particles could be present but not identified as such. Therefore, the difference in mixed-phase fractions between cloud and precipitation measurements is highly uncertain. However, the mixed-phase fraction of both shows a slight increase with distance to the coast.

## 4 Summary and Conclusions

Here we presented an overview of liquid and mixed-phase clouds and precipitation for marine boundary layer clouds over the WNAO. We have shown that the utilization of multiple phase spaces of microphysical parameters provides access to a classification of cloud and precipitation, which were partitioned into liquid and mixed-phase. With this classification, we provided an overview of all cloud measurements during the ACTIVATE campaign deployments in winter (February-March 2020, January-April 2021, and November 2021-March 2022) and summer (August-September 2020, May-June 2021, and May-June 2022). Vertical profiles of liquid and mixed-phase cloud properties were shown separately and the spatial distribution and frequency of clouds and precipitation were discussed. The findings are listed below:

– LWC of clouds increases with altitude in the MBL and shows large seasonal variability. In addition, the average number of clouds encountered per research flight is higher in winter. Overall, our analysis indicates that more overcast conditions occurred during the winter deployments, whereas summer was dominated by an increased number of broken cloud scenes.

– The altitude distribution of MBL clouds shows a maxima near 1.3 km in winter and 0.5 km in summer, and is consistent with higher satellite-based cloud top height from MODIS. In winter the majority of MBL clouds were observed below 0°C and contained mixed-phase and ice clouds. If ice particles were seen in the winter seasons, they were mostly composed of graupel and were observed for temperatures up to 3.5°C. Non-spherical particles without hexagonal structure occurred at temperatures above 10°C during summer, suggesting the presence of scarce and large bioaerosol.

– Vertical profiles of pure liquid clouds show higher $N_{liquid}$ and lower $ED_{liquid}$ for MBL clouds in winter compared to summer. The LWC values and profiles are comparable in both seasons. Therefore, the anti-correlation of $N_{liquid}$ and $ED_{liquid}$ between seasons shows a less efficient collision-coalescence process in winter, leading to a higher suppression of precipitation for pure liquid clouds during winter.

– Vertical profiles of mixed-phase clouds show a decrease in the liquid parameters $N_{liquid}$ and LWC, and a simultaneous increase in IWC, $N_{ice}$, and $ED_{ice}$ above 1.5 km. This shows enhanced glaciation and can be explained by the onset of the WBF process.

– The particle size distribution of mixed-phase clouds shows a higher concentration of precipitation particles >100 $\mu$m up to two orders of magnitude compared to pure liquid clouds. This can be explained by the onset of the ice phase,

processes such as WBF, coalescence, riming and aggregation, and secondary ice formation, together with dynamic influences. Therefore, the initiation of the ice phase promotes stronger precipitation.

– The frequency of precipitation is higher in winter than in summer. The spatial distribution shows that in winter most of the cloud measurements are associated with precipitation in the same area, while the frequency of precipitation sampling coincides less frequently with cloud measurements during summer. In addition, the mixed-phase fraction of clouds and precipitation measurements show a latitudinal decline with highest fractions north of 37.5˚N.

The results presented in this study provide an overview of the cloud data measured during ACTIVATE and show how the data can be classified. Also, we demonstrated that the data collected from the statistical sampling strategy can be used to derive macro- as well as microphysical cloud properties from in-situ data. As the flight strategy was statistically oriented, the wealth of cloud and precipitation data will help to develop parameterizations for climate and weather models. Lastly, the dataset is particularly well suited for investigating the processes that give rise to liquid and mixed-phase clouds, ice and precipitation, which are generally associated with cold-air outbreaks.

*Data availability.* The ACTIVATE data are available at http://doi.org/10.5067/SUBORBITAL/ACTIVATE/DATA001

*Author contributions.* S.K. conducted the analysis and wrote the manuscript. C.V. advised the study and provided feedback on the manuscript. D.P. wrote the last paragraph of 2.5.1.. S.K., V.H., St.K and C.R. participated in instrument calibration. B.A., R.M., L.Z. and A.S. participated in mission planning. R.F., A.J.S. and M.S. conducted the weather forecast. E.C, R.F., A.J.S., R.M., M.S., J.W.H., T.S. and A.S. participated in strategic flight planning. S.K., E.C., R.M., K.T., C.R, E.W., L.Z., M.S., J.W.H., T.S and A.S. participated in mission operation. G.C. and M.S. conducted the data management. All authors commented on the manuscript.

*Competing interests.* The authors declare that they have no conflict of interest.

*Acknowledgements.* The work was supported by ACTIVATE, a NASA Earth Venture Suborbital-3 (EVS-3) investigation funded by NASA's Earth Science Division and managed through the Earth System Science Pathfinder Program Office. C.V, S.K. and St.K. were funded by the Deutsche Forschungsgemeinschaft (DFG, German Research Foundation) – TRR 301 – Project-ID 428312742 and the SPP 1294 HALO under contract VO 1504/7-1 and VO 1504/9-1. University of Arizona investigators were funded by NASA grant 80NSSC19K0442.

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
