# Peer review of "Overview and statistical analysis of boundary layer clouds and precipitation over the western North-Atlantic Ocean"

_EGUsphere, 2023_

## Referee Comment (RC1)

**Review of "Overview and statistical analysis of boundary layer clouds and precipitation over the western North-Atlantic Ocean" by Kirschler et al. (egusphere-2023-898)**

**General comments:**

This study provides an overview of the in-situ cloud sampling dataset collected during the NASA ACTIVATE mission over the western North-Atlantic Ocean. The ACTIVATE team did a great job of collecting comprehensive observations and the authors have compiled these observations in a way that sets the stage well for further examination. This study compares measurements of cloud and precipitation properties across summer and winter months from three deployment years. The dataset is described with extensive descriptions of cloud probes and their operational characteristics. A merged particle size distribution is created using two data from cloud probes and a hydrometeor number-size phase space is used to classify the observations into aerosol, cloud, and precipitation samples. The methodology is described with great detail, which is appreciated. The discussion of vertical profiles and spatial distribution of cloud/precipitation properties presents findings in line with existing knowledge. The novel aspects of the study include the rich in-situ dataset and a comparison of seasonal trends in cloud properties.

In certain instances, the paper could benefit from better organization or minor clarifications. The introduction focuses on aerosol variability a bit too much given that aerosols are not the focus of the analysis. Some interesting discussions could be added, and conclusions could be rephrased slightly to separate direct observations from inferences made based on these observations. However, these are minor details and I recommend the paper for publication with minor revisions. Specific comments are provided below along with a list of technical edits for the authors' consideration.

**Specific comments:**

Line 1 and 15: Please specify what "large variability" is referring to?

Line 19: I don't see how the phase space helps understand cloud formation. As I interpreted the discussion, the phase space provided thresholds to separate cloud samples from precipitation/aerosol sampling and to shed light on hydrometeor phase.

Line 25-38: This paragraph has some very old references which can be supplemented with more recent work. For example, quantitative cloud radiative effect estimates could be provided?

Line 39: is there a reference associated with this statement?

Line 40: Please specify what "aerosol variability" is referring to. I find text that might be relevant in lines 49-55 which could be placed with this statement for clarity.

Line 61: There is considerable emphasis on aerosol variability. Aerosol conditions are discussed only to be followed by a statement that updraft speeds predominantly dictate the wintertime cloud properties (or at least number concentration). The paper has little to no analysis or mention of aerosol observations. I suggest the introduction be steered from aerosol variability toward the discussion of cloud processes which are explained later in the sub-sections. The extended discussion of previous knowledge of cloud

processes within the results seems a bit misplaced. Perhaps the aerosol discussion can be moved to a discussion section if the authors still wish to discuss these aspects?

Line 95: If the CAPS system was also available and collecting data, was there a reason to choose FCDP data for this study? Would the results presented in this study be affected if the CAPS data were used instead? Since these probes sample droplets in similar size ranges, these data could be compared. I would assume the conclusions of the study are not sensitive to the choice of the probe but that would be good to confirm and report here.

Line 115: I appreciate the mention and quantification of the measurement uncertainties. Do the uncertainties affect the interpretation of the results? A comment on that would be helpful.

Line 117: The term "The FCDP and 2D-S combination probe" seems strange. I would suggest using a term that highlights that this is a combined dataset instead. As it is presently stated, it seems a new combined probe was created when in fact data from two probes were merged to create a combined size distribution.

Line 143: This information could be placed with "we assume all small particles <100 $\mu$ m detected by the FCDP and the 2D-S to be liquid, as there is no other information available". The current organization is confusing as the explanation is placed separately.

Line 149: quantitative estimate in place of "vast majority"?

Figure 2:

1. I'm assuming the axis limits on panels (b) and (c) were chosen for consistency with panel 2a. It could be argued that that is not important. The three panels show three different 2-D phase spaces. The panels would be better interpreted if they have axis limits corresponding to the data plotted within them.
2. I don't see the utility (or any discussion) of the counts presented in panels (b) and (c). I suggest coloring these panels according to N_liquid and/or ED_liquid for the following reasons:
   o The discussion of Fig. 2a argues the 2-D phase space of ED_liquid and N_liquid is necessary to determine the thresholds used to determine in/out of cloud and precipitation samples.
   o Independent thresholds for N_liquid to determine cloud samples and ED_liquid to determine precipitation samples have consistently been used by past studies. These studies have used thresholds with similar values to those determined based on Fig. 2a. (in fact, the current N_liquid threshold is partly influenced by the instrument detection limit).
   o Coloring by N_liquid and/or ED_liquid would likely help reinforce the choice of thresholds.
3. It is not obvious how the threshold of ED_liquid > 60 um was chosen to represent precipitation samples based on Fig. 2a. On first glance, there seem to be no significant differences in N_liquid counts for ED_liquid between 40 um to 60 um.

Line 179: A combination of N_liquid and LWC thresholds to define cloud samples has been used previously. A reference to the past studies and contrasting the current thresholds with past work would be useful (Wood, 2005 (https://doi.org/10.1175/JAS3529.1) or Gupta et al., 2021 (https://doi.org/10.5194/acp-21-4615-2021 - this is a non-exhaustive list).

Line 213: It would be nice to include statistics for ED_liquid in Table 1. That would also allow including these quantities while comparing the dominant particle modes during the three summer deployments.

Line 213: Given the large LWC differences between winter deployments, I'm wondering why winter deployments were not compared similar to comparisons between summer deployments and summer 2020 versus winter 2020.

Line 254: Suggest changing "thus yielding stronger suppression of precipitation during winter as compared to summer" to "thus yielding lower concentrations for droplets larger than __ um during the winter" and referring to figure 6d. This is what was observed based on the measurements instead of observing precipitation suppression directly.

Line 276: "Mixing processes…". Was this observed/explained by Tornow et al., 2022? As currently phrased, it seems this was determined based on current analysis. Suggest the sentence be edited or placed after the next one if explained by Tornow et al. 2022.

Line 281: From Fig. 7h, it seems the ice cloud size distribution exceeds the liquid cloud size distribution after 200 um instead of 100 um. This is probably due to the sample volume of the 2D-S, but I suggest either changing to 200 um or clarifying the selection of 100 um here.

Line 297: The phrasing is confusing. I am assuming you mean something like this - "During summer, the cloud deck is characterized by more cloud-free areas and the width of the clouds is reduced with less frequent cloud measurements compared to the winter seasons."?

Line 299: Figure 8 presents a good case to compare cloud-free areas, but I wouldn't imagine this is a good way to compare seasonal cloud width. For direct comparisons of cloud width, wouldn't it be better to compare the average duration of cloud transects during the seasons?

Figure 10: I don't see a discussion of panels c and f? These panels interesting features and spatial patterns. Maybe I'm missing something, but I'm confused by the observation of higher fractions of mixed-phase precipitation (Fig. 10f) versus the fraction of mixed phase clouds (Fig. 10c)?

Line 349: "In addition, the average cloud encountered per research flight is higher in winter." I find this statement confusing. Do you mean you sampled more clouds during the winter or sampled clouds for a greater duration during the winter? I am assuming the latter.

Line 356: Suggest adding the rationale behind the observation made here.

Line 359: The vertical profiles do not show the onset of WBF process. The onset is inferred based on the observations that are consistent with changes expected in association with the WBF process. Suggest rephrasing. Similar adjustments would be helpful for the following paragraph.

Line 367: "consists of a larger mixed-phase fraction compared to clouds observed in the summer"?

Line 368: "while precipitation coincides less frequently with cloud measurements during summer". This phrasing seems a bit confusing since precipitation would generally be associated with a cloud which was not sampled. The frequency of precipitation was likely higher due to the sampling altitude? Suggest changing this to "frequency of precipitation sampling coincides less frequently with cloud measurements during summer" or clarify if this interpretation is incorrect.

**Technical corrections:**

Line 21: "products"?

Line 29: "higher cooling rates by low-level clouds"?

Line 53: "show"?

Line 70: "Cloud, aerosol, trace gas"?

Line 140: I believe you mean $D_i$ and $N_i$ in this sentence instead of D and N?

Line 159: It would be useful to explicitly mention the color of each corresponding region within the text.

Line 262: "we assume particles with sizes smaller than 100 $\mu$ m to be liquid" reads better.

Figure 8: Suggest changing the color of the symbols to represent liquid or mixed phase. It is nearly impossible to see the difference for the darker blue colors for winters 2021 and 2022. Might also be more informative to include the number of flights or flight seconds for each season in the legend.

Line 346: "cloud properties" instead of "clouds"?

Line 349: "indicates"?

Line 353: "contained"?

Line 362: "size distribution of mixed-phase cloud droplets" unless you mean the size distribution of clouds themselves.

Line 365: promotes "stronger precipitation" instead of "precipitation"?

Line 371: "oriented"

Figure 3,4,5: Please consider adding labels to subplots within these figures.

---

## Author Comment (AC1)

**Response to Reviewers of "Overview and statistical analysis of boundary layer clouds and precipitation over the western North-Atlantic Ocean" by Kirschler et al. (egusphere-2023-898)**

**Reviewer #1:**

**General comments:**

This study provides an overview of the in-situ cloud sampling dataset collected during the NASA ACTIVATE mission over the western North-Atlantic Ocean. The ACTIVATE team did a great job of collecting comprehensive observations and the authors have compiled these observations in a way that sets the stage well for further examination. This study compares measurements of cloud and precipitation properties across summer and winter months from three deployment years. The dataset is described with extensive descriptions of cloud probes and their operational characteristics. A merged particle size distribution is created using two data from cloud probes and a hydrometeor number-size phase space is used to classify the observations into aerosol, cloud, and precipitation samples. The methodology is described with great detail, which is appreciated. The discussion of vertical profiles and spatial distribution of cloud/precipitation properties presents findings in line with existing knowledge. The novel aspects of the study include the rich in-situ dataset and a comparison of seasonal trends in cloud properties.

In certain instances, the paper could benefit from better organization or minor clarifications. The introduction focuses on aerosol variability a bit too much given that aerosols are not the focus of the analysis. Some interesting discussions could be added, and conclusions could be rephrased slightly to separate direct observations from inferences made based on these observations. However, these are minor details and I recommend the paper for publication with minor revisions. Specific comments are provided below along with a list of technical edits for the authors' consideration.

*We thank the reviewer for the insightful summary and the positive evaluation of the manuscript. The comments have improved the quality of the manuscript.*

**Specific comments:**

Line 1 and 15: Please specify what "large variability" is referring to?

*In line 1 the large variability refers to macro- and microphysical properties of clouds like cloud cover, thickness, cloud particle number concentrations, effective diameter, a spectrum between a uni- or bimodal size distribution, thermodynamical phase of the particles, etc. For clarification we have change the text to: "Due to their fast evolution and large natural variability in macro- and microphysical properties,…"*
*In line 15 we refrain from mentioning the bandwidth as the statement refers to relative values of the flight cloud cover. Since the flight cloud cover is a statistically derived parameter and cannot be generalized to other in-situ field campaigns, we think noting the quantitative values is not useful and can be looked up in more detail in the results section, where the necessary context is provided.*

Line 19: I don't see how the phase space helps understand cloud formation. As I interpreted the discussion, the phase space provided thresholds to separate cloud samples from precipitation/aerosol sampling and to shed light on hydrometeor phase.

*We thank the reviewer for pointing this out. Indeed, the underlying premise of this statement is that one can infer cloud formation from knowledge of cloud microphysical properties, e.g. hydrometeor phase, which we agree cannot be generalized. We have therefore changed the text to: "The evaluation of boundary layer clouds and precipitation in the $N_{liquid}$-ED phase space sheds light on liquid, mixed-phase, and ice cloud properties and helps to categorize the cloud data."*

Line 25-38: This paragraph has some very old references which can be supplemented with more recent work. For example, quantitative cloud radiative effect estimates could be provided?

We thank the reviewer for pointing that out and added the references:

Gettelman, A. and Sherwood, S. C.: Processes Responsible for Cloud Feedback, Current Climate Change Reports, 2, 179–189, https://doi.org/10.1007/s40641-016-0052-8, 2016.

Stephens, G. L., Li, J., Wild, M., Clayson, C. A., Loeb, N., Kato, S., L'Ecuyer, T., Stackhouse, P. W., Lebsock, M., and Andrews, T.: An Update on Earth's Energy Balance in Light of the Latest Global Observations, Nature Geoscience, 5, 691–696, https://doi.org/10.1038/ngeo1580, 2012.

Henderson, D. S., L'Ecuyer, T., Stephens, G., Partain, P., and Sekiguchi, M.: A Multisensor Perspective on the Radiative Impacts of Clouds and Aerosols, Journal of Applied Meteorology and Climatology, 52, 853–871, https://doi.org/10.1175/JAMC-D-12-025.1, 2013.

Wang, M., Su, J., Xu, Y., Han, X., Peng, N., and Ge, J.: Radiative Contributions of Different Cloud Types to Regional Energy Budget over the SACOL Site, Climate Dynamics, https://doi.org/10.1007/s00382-022-06651-0, 2023.

Line 39: is there a reference associated with this statement?

We thank the reviewer for catching the missing reference and changed the text accordingly to: *"The western North-Atlantic Ocean (WNAO) is one of the regions where the CMIP6 multi-model mean surface temperature significantly departs from observations (Bock et al., 2020)."*

Line 40: Please specify what "aerosol variability" is referring to. I find text that might be relevant in lines 49-55 which could be placed with this statement for clarity.

The term "aerosol variability" refers to changes in the sources, species and seasonal abundance of aerosols. As already noted by the reviewer, the "aerosol variability" refers to the references, which are given in lines 49 - 55. Sorooshian et al. (2020) provide an overview of previous studies on the main sources of aerosols (marine, urban, biogenic, dust and smoke) in the WNAO. Detailed information on the seasonal changes in the species and abundance of aerosols can be found in Corral et al. (2021). For clarification we have changed the text to: "*The WNAO has a broad spectrum of aerosol sources, species and abundances (Sorooshian et al., 2020; Corral et al., 2021). In addition, the WNAO has a wide range of meteorological conditions with mainly low shallow cumulus clouds …"*

Line 61: There is considerable emphasis on aerosol variability. Aerosol conditions are discussed only to be followed by a statement that updraft speeds predominantly dictate the wintertime cloud properties (or at least number concentration). The paper has little to no analysis or mention of aerosol observations. I suggest the introduction be steered from aerosol variability toward the discussion of cloud processes which are explained later in the sub-sections. The extended discussion of previous knowledge of cloud processes within the results seems a bit misplaced. Perhaps the aerosol discussion can be moved to a discussion section if the authors still wish to discuss these aspects?

We thank the reviewer for his suggestion. We restructured the paragraph and added a discussion of previous knowledge of cloud processes.

Line 95: If the CAPS system was also available and collecting data, was there a reason to choose FCDP data for this study? Would the results presented in this study be affected if the CAPS data were used

instead? Since these probes sample droplets in similar size ranges, these data could be compared. I would assume the conclusions of the study are not sensitive to the choice of the probe but that would be good to confirm and report here.

We chose the FCDP because of its additional features over the CAPS system's CDP. It provides a complete per-particle stored information system that allows the application of several correction algorithms such as a waveform symmetry filter and a transit time filter for coincidence correction. The FCDP has special arm tips to minimize the effects of shattering and the ability to perform inter-arrival time analysis to quantify and correct for possible shattering. In addition, a detailed laboratory characterization of the FCDP was performed to quantify the later reported uncertainty estimates that were not available for the CDP.

Line 115: I appreciate the mention and quantification of the measurement uncertainties. Do the uncertainties affect the interpretation of the results? A comment on that would be helpful.

We thank the reviewer for this question. The uncertainties given are for per second measurements of FCDP and 2D-S. In Section 2.2, the uncertainties for the combination of these probes are given for cloud and precipitation measurements, again on a per second basis. In this paper we follow a statistical approach using cloud events. While the measurement uncertainties remain as stated for cloud events on a per second basis, the longer a cloud event lasts, the lower the measurement uncertainty will be. In addition, the microphysical parameters within a cloud event have a large natural variability. However, all our results based on N/ED/LWC are determined by comparison and are therefore relative statements. We use a large number of cloud events where the within-cloud variability (which can be inferred from the given standard deviations) is mostly larger than the measurement uncertainties. Therefore, our interpretation of the results should not be affected by the measurement uncertainties.

Line 117: The term "The FCDP and 2D-S combination probe" seems strange. I would suggest using a term that highlights that this is a combined dataset instead. As it is presently stated, it seems a new combined probe was created when in fact data from two probes were merged to create a combined size distribution.

We thank the reviewer for highlighting the wording here. The FCDP and 2D-S combination is similar to the CDP and CIP combination in the CAPS system, which uses the same measurement principles of forward scattering and optical array probes. It is also mounted on the wing pot as a complementary probe in the same way as the CAPS. Therefore, we want to keep the wording here, as in the case of the CAPS system, we would likely state it in this way: "The CAPS measures particle size distributions...", too.

Line 143: This information could be placed with "we assume all small particles <100 μm detected by the FCDP and the 2D-S to be liquid, as there is no other information available". The current organization is confusing as the explanation is placed separately.

We thank the reviewer for this remark. We shifted the sentences *"The lower detection limit for ice is 100 μm in diameter, because we use a minimum number of 50 pixels for habit classification. Korolev and Sussman (2000) showed that the minimum pixel number should be between 20 and 60 for the separation of irregulars and spheres. An adequate number of pixels is necessary to extract shape information with sufficient accuracy. We use the particle area of the 2D images of each ice particle to derive the ice water content using the method of Baker and Lawson (2006) and the maximum diameter for sizing."* before the ED definition.

Line 149: quantitative estimate in place of "vast majority"?

In over 95% of the cases the FCDP measures more than 98% of all particles in cloud. We changed the text to: *"…are consistent with the FCDP uncertainties, since in over 95% of the in cloud measurements more than 98% of all cloud particles are measured by the FCDP."*

Figure 2:

1. I'm assuming the axis limits on panels (b) and (c) were chosen for consistency with panel 2a. It could be argued that that is not important. The three panels show three different 2-D phase spaces. The panels would be better interpreted if they have axis limits corresponding to the data plotted within them.

Indeed, we intended to be consistent with Panel 2a. We have changed the limits in panels (b) and (c).

2. I don't see the utility (or any discussion) of the counts presented in panels (b) and (c). I suggest coloring these panels according to N_liquid and/or ED_liquid for the following reasons:

- The discussion of Fig. 2a argues the 2-D phase space of ED_liquid and N_liquid is necessary to determine the thresholds used to determine in/out of cloud and precipitation samples.

- Independent thresholds for N_liquid to determine cloud samples and ED_liquid to determine precipitation samples have consistently been used by past studies. These studies have used thresholds with similar values to those determined based on Fig. 2a. (in fact, the current N_liquid threshold is partly influenced by the instrument detection limit).

- Coloring by N_liquid and/or ED_liquid would likely help reinforce the choice of thresholds.

We thank the reviewer for this suggestion and have applied it to Figure 2. Panel (b) is now color-coded with the mean $N_{liquid}$ values and panel (b) with the mean $ED_{liquid}$ values. The same changes have been made to Figure S1 in the Supplement. We changed the caption of Figure 2 accordingly to: *"The color code shows the number of seconds of cloud data and is the same for panels (a) and (d). The color code of panel (b)/(c) shows the mean $N_{liquid}$/$ED_{liquid}$ values."*
In addition, the text has been changed to: *"Figure 2b shows the phase space diagram for ice particles identified by the 2D-S with its lower size detection limit of 100 µm and represents precipitation of ice particles or snow. The color code shows that high $ED_{ice}$ and $N_{ice}$ values correlate with lower $ED_{liquid}$ values."*

3. It is not obvious how the threshold of ED_liquid > 60 um was chosen to represent precipitation samples based on Fig. 2a. On first glance, there seem to be no significant differences in N_liquid counts for ED_liquid between 40 um to 60 um.

We thank the reviewer for pointing this out, and we tested the thresholds at 40 and 50 µm, which changed the number of cloud events but did not change the results in Section 3. We chose 60 µm as the threshold instead of 40/50 µm to minimize the classification of possible background measurements where a single large particle could trigger the threshold into the precipitation category.

Line 179: A combination of N_liquid and LWC thresholds to define cloud samples has been used previously. A reference to the past studies and contrasting the current thresholds with past work would be useful (Wood, 2005 (https://doi.org/10.1175/JAS3529.1) or Gupta et al., 2021 (https://doi.org/10.5194/acp-21- 4615-2021 - this is a non-exhaustive list).

We thank the reviewer for providing the references. Gupta et al., (2021) used the combination of $N_{liquid}$ and LWC thresholds similar to our analysis. The study by Wood (2005) uses a case-by-case distinction between only LWC or only $N_{liquid}$ as the threshold for in cloud seconds, rather than the combination of both for each measurement second. The latter definition is dependent on $N_{liquid}$, similar to our analysis and uses the same threshold values for $N_{liquid}$ and LWC. It is convincing that both studies contain stratocumulus clouds and share or partly share our threshold values. We included the references to these past studies and changed the text to: "*Given the disadvantages of using a single LWC-based threshold, we add a threshold for $N_{liquid}$ similar to previous studies (e.g. Gupta et al., 2021; Wood, 2005), because it provides a better differentiation between in-cloud and out-of-cloud situations and avoids a misclassification of precipitation.*"

Line 213: It would be nice to include statistics for ED_liquid in Table 1. That would also allow including these quantities while comparing the dominant particle modes during the three summer deployments.

We thank the reviewer for this suggestion and have added $ED_{liquid}$ to Table 1. We have changed the text to: "*The timeframe varies between the summer seasons. LWC, $N_{liquid}$ and $ED_{liquid}$ of summer 2020 suggests a particle size distribution with a dominant small particle mode below 50 μm and less large particles in late summer.*"

Line 213: Given the large LWC differences between winter deployments, I'm wondering why winter deployments were not compared similar to comparisons between summer deployments and summer 2020 versus winter 2020.

The analysis of dynamical effects on cloud number concentration was performed by Kirschler et al. (2022) for 2020 only. Therefore, we did not want to generalize their results to 2021 and 2022. The winter deployments have similar time frames, temperature ranges, and their LWC differences could be due to different meteorological or synoptic conditions, differences in updraft speeds, and aerosol abundances. A future study is planned to investigate the observed LWC differences with a detailed analysis of the mentioned meteorological parameters.

Line 254: Suggest changing "thus yielding stronger suppression of precipitation during winter as compared to summer" to "thus yielding lower concentrations for droplets larger than __ um during the winter" and referring to figure 6d. This is what was observed based on the measurements instead of observing precipitation suppression directly.

We thank the reviewer for this comment and agree to be more specific here. We have changed the text to: "*The higher $N_{liquid}$ and smaller $ED_{liquid}$ of liquid marine boundary layer clouds increase cloud lifetime and suppress processes leading to precipitation and thus yielding lower concentrations for droplets larger than 40 μm during the winter as compared to summer (Albrecht, 1989; Freud et al., 2011; Freud and Rosenfeld, 2012; Braga et al., 2021b).*"

Line 276: "Mixing processes…". Was this observed/explained by Tornow et al., 2022? As currently phrased, it seems this was determined based on current analysis. Suggest the sentence be edited or placed after the next one if explained by Tornow et al. 2022.

We thank the reviewer for this comment. This sentence is not a reference to Tornow et al. (2022). Here we state our logical chain with the premises on how mixing processes should affect the cloud microphysical properties.

Line 281: From Fig. 7h, it seems the ice cloud size distribution exceeds the liquid cloud size distribution after 200 um instead of 100 um. This is probably due to the sample volume of the 2D-S, but I suggest either changing to 200 um or clarifying the selection of 100 um here.

We thank the reviewer for pointing this out. In this statement we use the term "large particles" to include both liquid and ice. Here, we want to compare the total cloud number concentrations of these large particles greater than 100 μm with those of liquid clouds. A minimum of 50 pixels is used for ice classification, see section 2.2. While some habits such as columns and dendrites are easy to detect, graupel has a spherical shape and is more likely to be misclassified as liquid with this number of pixels, and the classification works better with more pixels where the edges of the particles are better resolved. Therefore, it is very likely that smaller ice particles than 100 μm are present in the observed clouds and that the liquid particle size distribution between 100 and 200 μm in Figure 7h does contain misclassified ice particles.

Line 297: The phrasing is confusing. I am assuming you mean something like this - "During summer, the cloud deck is characterized by more cloud-free areas and the width of the clouds is reduced with less frequent cloud measurements compared to the winter seasons."?

We thank the reviewer for this question. The more cloud-free areas are a relative statement and are determined by the average number of cloud events per research flight. The width of the clouds is approximated by the duration of the cloud event and the average Falcon HU-25 TAS during the campaign. Therefore, both statements refer to cloud events. This section 3.4. analyzes cloud measurements on a per second basis instead of cloud events. The purpose of this sentence is to show that both the analysis of cloud events and the per second measurements lead to the same conclusion, i.e. reduced cloud cover in summer. This is generally not true, since the duration depends on the cloud type and is an attribution of the measured seconds to a specific cloud. We hope that we have sufficiently addressed why this phrasing was used. To highlight the change to measurements on a per second basis we changed the text to : *Figure 8 shows the frequency of precipitation and cloud measurements on a per second basis in winter and summer 2020, 2021, and 2022, using all seconds that fall into the precipitation ($N_{liquid} < 10$ cm-3 and $ED_{liquid} > 60$ μm) or cloud ($N_{liquid} > 10$ cm-3 and LWC > 0.02 g m3) category."* In the beginning of Section 3.4.

Line 299: Figure 8 presents a good case to compare cloud-free areas, but I wouldn't imagine this is a good way to compare seasonal cloud width. For direct comparisons of cloud width, wouldn't it be better to compare the average duration of cloud transects during the seasons?

Exactly, Figure 8 does not necessarily determine cloud-free areas as well as the analysis of cloud width and the average number of cloud events per research flight in Section 3.1, see Table 1. However, Figure 8 provides valuable information on cloud type and precipitation that is only partially included in Section 3.1. In the analysis of Section 3.1, stratiform clouds are masked by the use of cloud events because there is no weighting of cloud events by their duration. For example, the summer of 2022 had a relatively high fraction of flights with stratiform cloud decks, which can be seen in the frequency. The fact that the average number of cloud events per RF in Table 1 is reflected in the cloud frequency in Table 8 assures us that the dominant cloud type was cumulus, which is consistent with the reports of the flight scientist onboard the Falcon HU-25. In addition, we have to use measurements on a per second basis here because we have not defined precipitation events and need the same norm to determine fractions correctly.

Figure 10: I don't see a discussion of panels c and f? These panels interesting features and spatial patterns. Maybe I'm missing something, but I'm confused by the observation of higher fractions of mixed-phase precipitation (Fig. 10f) versus the fraction of mixed phase clouds (Fig. 10c)?

We thank the reviewer for catching the missing discussion on panels c and f. We added the paragraph: "*The mixed-phase fraction of clouds and precipitation measurements in Figure 10c,f show a latitudinal decline with highest fractions north of 37.5˚N. It is defined as measurement seconds containing ice particles to all cloud or precipitation measurements in the regular grids, respectively. The mixed-phase fraction of the precipitation measurements is higher than that of the cloud measurements. This may be due to the lower detection limit for ice classification of the 2D-S. It is likely that the cloud measurements had a higher fraction of mixed-phase or ice clouds that could not be detected. In those cloud measurement seconds, no large ice particles were measured by the 2D-S. Thus, smaller ice particles could be present but not identified as such. Therefore, the difference in mixed-phase fractions between cloud and precipitation measurements is highly uncertain. However, the mixed-phase fraction of both shows a slight increase with distance to the coast*.*"*

Line 349: "In addition, the average cloud encountered per research flight is higher in winter." I find this statement confusing. Do you mean you sampled more clouds during the winter or sampled clouds for a greater duration during the winter? I am assuming the latter.

We thank the reviewer for pointing this out. We have changed the text to: "*In addition, the average number of clouds encountered per research flight is higher in winter.*"

Line 356: Suggest adding the rationale behind the observation made here.

We thank the reviewer for his suggestion and changed the text to: "*Therefore, the anti-correlation of $N_{liquid}$ and $ED_{liquid}$ between seasons shows a less efficient collision-coalescence process in winter, leading to a higher suppression of precipitation for pure liquid clouds during winter.*"

Line 359: The vertical profiles do not show the onset of WBF process. The onset is inferred based on the observations that are consistent with changes expected in association with the WBF process. Suggest rephrasing. Similar adjustments would be helpful for the following paragraph.

We thank the reviewer for pointing that out and changed the text to:" *Vertical profiles of mixed-phase clouds show a decrease in the liquid parameters $N_{liquid}$ and LWC, and a simultaneous increase in IWC, $N_{ice}$, and $ED_{ice}$ above 1.5 km. This shows enhanced glaciation and can be explained by the onset of the WBF process.*

*– The particle size distribution of mixed-phase clouds shows a higher concentration of precipitation particles >100 µm up to two orders of magnitude compared to pure liquid clouds. This can be explained by the onset of the ice phase, processes such as WBF, coalescence, riming and aggregation, and secondary ice formation, together with dynamic influences. Therefore, the initiation of the ice phase promotes stronger precipitation.*"

Line 367: "consists of a larger mixed-phase fraction compared to clouds observed in the summer"?

We thank the reviewer for pointing to that. Here, we do not compare mixed-phase fraction of winter measurements to summer, since very few mixed-phase clouds were observed in summer. Instead, we want to state that the precipitation measurements consist of a higher fraction of mixed-phase measurements than cloud measurements which may be due to the limitations of the 2D-S. We changed the sentences to: "*The frequency of precipitation is higher in winter than in summer. The spatial distribution shows that in winter most of the cloud measurements are associated with precipitation in the same area, while the frequency of precipitation sampling coincides less frequently*

*with cloud measurements during summer. In addition, the mixed-phase fraction of clouds and precipitation measurements show a latitudinal decline with highest fractions north of 37.5˚N."*

Line 368: "while precipitation coincides less frequently with cloud measurements during summer". This phrasing seems a bit confusing since precipitation would generally be associated with a cloud which was not sampled. The frequency of precipitation was likely higher due to the sampling altitude? Suggest changing this to "frequency of precipitation sampling coincides less frequently with cloud measurements during summer" or clarify if this interpretation is incorrect.

We thank the reviewer for this comment. Indeed, precipitation would generally be associated with a cloud which was not sampled. Due to the flight strategy, the statistical pattern of the HU-25, the leg distances below and in cloud are comparable. In fact, the flight scientist often tried to expand the in cloud legs, which would favor the frequency of cloud measurements. Therefore, a bias due to sampling altitudes is not likely. However, the wording is not specific enough. We have changed the text as suggested.

**Technical corrections:**

Line 21: "products"?

We thank the reviewer and changed the text accordingly.

Line 29: "higher cooling rates by low-level clouds"?

We thank the reviewer and changed the text accordingly.

Line 53: "show"?

We thank the reviewer and changed the text accordingly.

Line 70: "Cloud, aerosol, trace gas"?

We thank the reviewer and changed the text accordingly.

Line 140: I believe you mean $D_i$ and $N_i$ in this sentence instead of D and N?

We thank the reviewer and changed the text accordingly.

Line 159: It would be useful to explicitly mention the color of each corresponding region within the text.

We thank the reviewer and changed the text accordingly.

Line 262: "we assume particles with sizes smaller than 100 µm to be liquid" reads better.

We thank the reviewer and changed the text accordingly.

Figure 8: Suggest changing the color of the symbols to represent liquid or mixed phase. It is nearly impossible to see the difference for the darker blue colors for winters 2021 and 2022. Might also be more informative to include the number of flights or flight seconds for each season in the legend.

We thank the reviewer and changed the colors.

Line 346: "cloud properties" instead of "clouds"?

We thank the reviewer and changed the text accordingly.

Line 349: "indicates"?

We thank the reviewer and changed the text accordingly.

Line 353: "contained"?

We thank the reviewer and changed the text accordingly.

Line 362: "size distribution of mixed-phase cloud droplets" unless you mean the size distribution of clouds themselves.

We thank the reviewer and changed the text to: "*The particle size distribution of mixed-phase clouds…*"

Line 365: promotes "stronger precipitation" instead of "precipitation"?

We thank the reviewer and changed the text accordingly.

Line 371: "oriented" Figure 3,4,5: Please consider adding labels to subplots within these figures.

We thank the reviewer and changed the text accordingly. We also added labels to the subplots of Figure 4 and changed the references in the text.

**Reviewer #2:**

**General Comments:**

This manuscript uses ACTIVATE data, which spanned 3 summer and 3 winter seasons, that sampled over 17,000 cloud events using the FCDP and 2D-S in-situ probes. The authors found, consistent with many other previous studies, that both LWC and cloud droplet effective diameter increase with altitude inside clouds. Higher updraft speed in winter boundary layer clouds explains higher cloud droplet number concentrations despite lower CCN. A seasonal contrast in cloud type and coverage is noted: inter clouds are typically stratocumulus, aided by passing synoptic-scale low pressure/frontal systems, whereas summer clouds are more likely to be open cell cumulus. The strength of this paper lies in the thorough overview of the vast quantity of in-situ measurements of boundary layer clouds, whether liquid, mixed-phase or ice clouds, and quantifying their relative occurrence through the entire 3-year dataset.

This manuscript is very well written, easy to follow along, and comes to accurate and valuable conclusions that will be useful to the broader scientific community looking to investigate cloud properties in this region of the globe across many applications (LES and climate modeling, meteorological influences, etc.). The only area of the manuscript that, in my opinion, needs improvement is in the introduction/literature review: there were many broad-reaching statements that need to be supported (or could be better supported) by recent studies. Specific comments are noted below. I also noted a few areas later in the paper where the results of this work could be directly compared, namely with other similar suborbital campaigns. I think the authors should consider a follow-on study quantifying the role of meteorological influences to the observed cloud

properties, as the manuscript makes several references explaining why observed cloud properties occur but only provides temperature data for context. I think this manuscript is very clean in its current form and adding such analysis – in my opinion – is better off as its own separate analysis and manuscript.

Overall, this manuscript merits publication in EGUsphere. The majority of my comments are "minor" in nature, and the authors have a high degree of flexibility in how they choose to address them.

We thank the reviewer for the positive evaluation of our manuscript. We indeed plan a follow-on study quantifying the role of meteorological influences on the observed cloud properties, i.e. the synoptical patterns.

**Specific Comments:**

L25-38 (Paragraph 1): This paragraph flows well, but several of the statements need additional references, especially more recent references that have addressed or provided new evidence for some of the statements.

We thank the reviewer for pointing that out and added the references:

Gettelman, A. and Sherwood, S. C.: Processes Responsible for Cloud Feedback, Current Climate Change Reports, 2, 179–189, https://doi.org/10.1007/s40641-016-0052-8, 2016.

Stephens, G. L., Li, J., Wild, M., Clayson, C. A., Loeb, N., Kato, S., L'Ecuyer, T., Stackhouse, P. W., Lebsock, M., and Andrews, T.: An Update on Earth's Energy Balance in Light of the Latest Global Observations, Nature Geoscience, 5, 691–696, https://doi.org/10.1038/ngeo1580, 2012.

Henderson, D. S., L'Ecuyer, T., Stephens, G., Partain, P., and Sekiguchi, M.: A Multisensor Perspective on the Radiative Impacts of Clouds and Aerosols, Journal of Applied Meteorology and Climatology, 52, 853–871, https://doi.org/10.1175/JAMC-D-12-025.1, 2013.
Wang, M., Su, J., Xu, Y., Han, X., Peng, N., and Ge, J.: Radiative Contributions of Different Cloud Types to Regional Energy Budget over the SACOL Site, Climate Dynamics, https://doi.org/10.1007/s00382-022-06651-0, 2023.

L29: You can just say "Weather systems", drop "Meteorological".

We thank the reviewer and changed the text accordingly.

L30: "... can induce ice nucleation or the formation of precipitation" reference(s) for this?

We thank the reviewer for pointing that out and added the references:

Naud, C. M. and Kahn, B. H.: Thermodynamic Phase and Ice Cloud Properties in Northern Hemisphere Winter Extratropical Cyclones Observed by Aqua AIRS, Journal of Applied Meteorology and Climatology, 54, 2283–2303, https://doi.org/10.1175/JAMC-D-15-0045.1, 2015.

Painemal, D., Chellappan, S., Smith Jr., W. L., Spangenberg, D., Park, J. M., Ackerman, A., Chen, J., Crosbie, E., Ferrare, R., Hair, J., Kirschler, S., Li, X.-Y., McComiskey, A., Moore, R. H., Sanchez, K., Sorooshian, A., Tornow, F., Voigt, C., Wang, H., Winstead, E., Zeng, X., Ziemba, L., and Zuidema, P.: Wintertime Synoptic Patterns of Midlatitude Boundary Layer Clouds Over the Western North Atlantic: Climatology and Insights From In Situ ACTIVATE Observations, Journal of Geophysical Research: Atmospheres, 128, e2022JD037 725, https://doi.org/10.1029/2022JD037725, 2023.

L31: "...due to the fast evolution and large variability of clouds..." again, some references for this statement showing this in CMIP6 (or previous versions of CMIP) would be good here.

We thank the reviewer for this comment. In the first part of the sentence, we refer to real clouds, not to their representation in models. However, we have added references to other groups that have analyzed cloud representation in CMIP models (see next comment below).

L32 "... the representation of clouds in climate models remains a challenge" how have other modeling groups found this challenging? I think you can expand this and make it stronger by adding another 1-2 sentences with additional supporting references.

We thank the reviewer or this question and have changed the text with additional references to: "*Hence, due to the fast evolution and large natural variability of clouds, the representation of clouds in climate models remains a challenge (Mülmenstädt and Feingold, 2018). The multimodel net cloud feedback in Coupled Model Intercomparison Project Phase 5 (CMIP5) models ranges from -0.13 to 1.24 W m-2 K-1 (Ceppi et al., 2017) and shows a larger range in CMIP6 models with an increase in their mean values from 0.09 to 0.21 W m-2 K-1 due to a decrease in low cloud coverage (Zelinka et al., 2020). Cesana et al. (2022) show that the reflected shortwave solar radiation is still underestimated in CMIP6 models compared to satellite observations in the Southern Ocean*."

L37: CMIP6 showed a large intermodel spread of what exactly? Please clarify.

We thank the reviewer for pointing this out and changed the text to: "… *CMIP6 shows larger intermodel spread in effective climate sensitivity than CMIP5 (Bock et al., 2020)*."

L39-40: "... significantly departs from observations." this is a strong statement, and needs supporting references.

We thank the reviewer for catching the missing reference and changed the text accordingly to: *"The western North-Atlantic Ocean (WNAO) is one of the regions where the CMIP6 multi-model mean surface temperature significantly departs from observations (Bock et al., 2020)."*

L42: Suggested re-write: "This provides ideal conditions" --> "... and frontal systems (Field et al., 2017a), providing ideal conditions for..."

We thank the reviewer for this suggestion, but want to stick to short sentences for readability.

L44-48: Were there other studies done to support the findings of X.-Y. Li & F. Tornow?

Yes, there are other studies supporting the findings of X-Y. Li & F. Tornow. We have added the references:

Abel, S. J., Boutle, I. A., Waite, K., Fox, S., Brown, P. R. A., Cotton, R., Lloyd, G., Choularton, T. W., and Bower, K. N.: The Role of Precipitation in Controlling the Transition from Stratocumulus to Cumulus Clouds in a Northern Hemisphere Cold-Air Outbreak, Journal of the Atmospheric Sciences, 74, 2293–2314, https://doi.org/10.1175/JAS-D-16-0362.1, 2017.

Goren, T., Feingold, G., Gryspeerdt, E., Kazil, J., Kretzschmar, J., Jia, H., and Quaas, J.: Projecting Stratocumulus Transitions on the Albedo—Cloud Fraction Relationship Reveals Linearity of Albedo to Droplet Concentrations, Geophysical Research Letters, 49, e2022GL101 169, https://doi.org/10.1029/2022GL101169, 2022

Wood, R., Bretherton, C. S., Leon, D., Clarke, A. D., Zuidema, P., Allen, G., and Coe, H.: An Aircraft Case Study of the Spatial Transition from Closed to Open Mesoscale Cellular Convection over the Southeast Pacific, Atmospheric Chemistry and Physics, 11, 2341–2370, https://doi.org/10.5194/acp-11-2341-2011, 2011

L60-62: The wording of this sentence is awkward. Do you mean to say "Altogether the WNAO experiences interesting and complex weather patterns, thus providing a natural laboratory to study shallow and broken cumulus clouds in a broad spectrum of aerosol and meteorological conditions."?

We thank the reviewer for pointing that out and changed the text to: "*Altogether the WNAO features interesting and complex weather patterns, providing a natural laboratory for studying shallow and broken cumulus clouds in a broad spectrum of aerosol and meteorological conditions*."

Figure 1: This is a very nice overview figure. Perhaps this question will be answered later as I continue reviewing this paper, but could you comment on the potential impact of the underlying Gulf Stream on potential cloud properties? The flight track locations mostly seem to take place across the primary area here, but I am curious if anyone has done work to show if the warm Gulf stream waters affect boundary layer cloud maintenance in the WNAO.

We consider the Gulf Stream to be a very important feature of the study region and thus affecting the marine boundary layer, i.e. during winter. Later we discuss how the Gulf Stream drives a deepening of the marine boundary layer with reference to Small et al. (2008) and Chelton et al. (2004). We are not aware of any study analyzing how the Gulf Stream affects the maintenance of boundary layer clouds, but we see the Gulf Stream as an important source of water vapor, enhanced surface heat fluxes, and consequently updraft velocities that favor cloud formation in this region.

L102: "15%/40%/45% respectively".

We thank the reviewer and changed the text accordingly.

L115: Are these calibration uncertainties true for all ACTIVATE flights? Please clarify. I would also add a reference here if these calibration uncertainties have been previously published.

The uncertainties apply to all ACTIVATE flights in cloud conditions. In section 2.2, the calibrated uncertainties for the combination of FCDP and 2D-S are given for cloud and precipitation measurements. These uncertainties are determined by laboratory experiments and have not been previously published.

Section 2.2: This section is very nicely detailed and written.

We are happy about the positive evaluation of the reviewer.

L141-142: Given LWC and IWC are integral parts of your study, you should show equations for both after Eq. (1).

We thank the reviewer for highlighting the importance of the equations and added them.

L149-150: I had a similar comment earlier, but are these "corresponding uncertainties" published elsewhere? If so, I would add a reference to that study here as well.

L172-176: You may find the following two studies interesting. Sinclair et al. (2021) showed using polarimetric and radar data how cloud top DSDs can be used to infer precipitation in conditions

where rainwater path (and hence liquid water path) might be quite low. Dzambo et al. (2021) used these same datasets to partition cloud path and rainwater path – Figure 2 in Section 4 shows that, when using RSP data to constrain cloud LWP, cloud water content is generally in the 0.05 g/m3 to ~0.4 g/m3 range, so mentioning that in-cloud LWC of ~0.02 g/m3 can exclude some precipitation makes for a very interesting comparison as both of those studies used ORACLES data (from the SE Atlantic, also a very aerosol-rich environment). Note for both studies that the results are for stratocumulus.

Sinclair, K., van Diedenhoven, B., Cairns, B., Alexandrov, M., Dzambo,

A. M., & L'Ecuyer, T. (2021). Inference of precipitation in warm stratiform clouds using remotely sensed observations of the cloud top droplet size distribution. Geophysical Research Letters, 48, e2021GL092547. https://doi. Org/10.1029/2021GL092547

General Comment: I am personally most familiar with the NASA ORACLES field campaign, but several other campaigns (e.g., CAMP2EX, NAAMES, SEAC4RS) have similar objectives to ACTIVATE but in different regions of the globe. I have added a few references worth considering to start (here and below), but I would recommend to the authors that they go through relevant and similar studies to this one on those campaigns and check to see how the results of this study compare to those. Ultimately, it will be very useful to the broader modeling communities how observed cloud properties vary by region, in addition to fortifying your own conclusions.

We thank the reviewer for pointing out these interesting studies. We have added Dzambo et al. (2021) as a reference for previous studies that use a combined threshold of LWC and $N_{liquid}$.

L209: "in summer" --> "during the summer"

We thank the reviewer and changed the text accordingly.

L211-212: "which could result in more frequent occurrence of stratiform cloud decks" this could indeed be true, but this raises the classic question regarding the relative roles of meteorological conditions versus background aerosol conditions in explaining the more frequent occurrence of stratiform. Can you comment on the role that atmospheric subsidence might play in maintaining stratiform? The works by Lee et al. (2009) and Jia et al. (2021, specifically Figs. 2 and 3) might be relevant here:

Lee, S. S., Donner, L. J., and Phillips, V. T. J.: Sensitivity of aerosol and cloud effects on radiation to cloud types: comparison between deep convective clouds and warm stratiform clouds over one-day period, Atmos. Chem. Phys., 9, 2555–2575, https://doi.org/10.5194/acp-9-2555-2009, 2009.

Jia, H., Ma, X., Yu, F. et al. Significant underestimation of radiative forcing by aerosol–cloud interactions derived from satellite-based methods. Nat Commun 12, 3649 (2021). https://doi.org/10.1038/s41467-021-23888-1

We thank the reviewer for drawing attention to this subject. Unfortunately, we do not know how subsidence affects marine boundary layer clouds in the WNAO during ACTIVATE. This is an interesting question that we should be able to investigate with the ACTIVATE data set., i.e. in the context that Jia et al. (2021) found an increase of cloud fraction up to 30% in the ACTIVATE region over the last two decades. In general, we think that the Gulf Stream is an important factor in this region and drives the dynamics, especially in winter. This in turn is an important component of cloud formation.

L227: This is beyond the scope of your study, but it would be really interesting to see a follow-on study to this work diving deeper into the role of meteorology (namely, inversion strength, subsidence strength, etc.) has on the cloud properties you observe here. Perhaps partition the results you show in Figure 4 into estimated inversion strength and a second variable (relative humidity or subsidence strength) similar to Figure 3 in Douglas and L'Ecuyer (2020):

Douglas, A. and L'Ecuyer, T.: Quantifying cloud adjustments and the radiative forcing due to aerosol–cloud interactions in satellite observations of warm marine clouds, Atmos. Chem. Phys., 20, 6225–6241, https://doi.org/10.5194/acp-20-6225-2020, 2020.

We thank the reviewer for providing the reference. Indeed, we are planning a follow-up study to quantify the role of meteorological influences on the observed cloud properties.

L240-242: This is very interesting. Figure 5 shows mixed-phase conditions quite clearly, relative to graupel conditions.

L253-255: Do you mean to say your results provide evidence here of precipitation suppression?

Our analysis of pure liquid clouds in Section 3.2 shows that higher Nliquid, lower EDliquid, and higher number concentration of droplets larger than 40 µm are measured during winter. We have a large number of cloud events classified as liquid and the differences in the particle size distribution (Figure 6d) are statistically relevant. This leads us to conclude that precipitation formation is suppressed in winter liquid clouds compared to summer liquid clouds. Since the suppression of precipitation is already stated in the first part of the sentence, and to be more precise, we have changed the text to: "*The higher $N_{liquid}$ and smaller $ED_{liquid}$ of liquid marine boundary layer clouds increase cloud lifetime and suppress processes leading to precipitation and thus yielding lower concentrations for droplets larger than 40 µm during the winter as compared to summer (Albrecht, 1989; Freud et al., 2011; Freud and Rosenfeld, 2012; Braga et al., 2021b)."*

L280: I agree with the conclusions in this paragraph.

Figure 8: To be clear, the frequency of precipitation implies the fraction (percent) of cloud events that had precipitation? Based on the frequency % of these results, some readers might assume that nearly every winter 2021 cloud event had observed precipitation.

We thank the reviewer for this question. We do not analyze cloud events in Section 3.4. Here we analyze precipitation and cloud measurements on a per second basis. We have not defined precipitation events. Cloud events provides mean values over their duration and we did not weight the cloud events with their duration. We have to use the same norm for assessing the frequency of precipitation in relation to cloud correctly. Therefore, we switched from cloud events to cloud and precipitation measurements on a per second basis. However, your observation is correct: a high fraction of cloud measurement correlate with precipitation measurements during winter which is not true for summer and discussed in the second part of Section 3.4. The flight strategy utilizes a stairstepping fashion which is a statistical approach and repeats a series of flight levels, see Dadashazar et al. (2022) and Sorooshian et al. (2019) for more details. To highlight the usage of per second measurements we have changed the text to: "*Figure 8 shows the frequency of precipitation and cloud measurements on a per second basis in winter and summer 2020, 2021, and 2022, using all seconds that fall into the precipitation ($N_{liquid}$ < 10 cm-3 and $ED_{liquid}$ > 60 µm) or cloud ($N_{liquid}$ > 10 cm-3 and LWC > 0.02 g m3) category."*

L302-303: Another idea you could pursue is investigating/applying the work of the aforementioned Sinclair et al. (2021) study to explore cloud top DSDs during ACTIVATE, and if enough data are available, could "bridge" the in-situ results to relevant satellite studies of the area. Validating satellite measurements with remote sensing instrumentation isn't an easy task, however...

We think this is an excellent idea and there are plans to compare ACTIVATE in-situ data with RSP and HSRL-2 data. There have been several underflights of orbiting satellites such as CALIPSO that could be included in this comparison. In a later step they could be compared with geostationary satellites.

L321: This could be a good area to add discussion about the role of the Gulf Stream, and any seasonal variability it has, on surface heat/moisture fluxes & subsequent influence on cloud formation.

We thank the reviewer for this suggestion and added the sentence: "*The ACTIVATE flights were mostly conducted over the Gulf Stream, and the increased frequency of cloud measurements in winter correlates with high surface latent and sensible heat fluxes driven by the Gulf Stream, which peak in winter and are considerably lower in summer (Painemal et al., 2021)."*

Figure 10: I really like this visualization method – very clever and informative given the vast amount of data you have.

We thank the reviewer for his positive feedback.

L335: This study is very clean and thorough in quantifying/assessing cloud properties observed throughout the ACTIVATE campaign, and I think as readers go through this work, they will likely have some good ideas for how to carry this work forward – a hallmark of a very well written manuscript. In my opinion, assessing the role of meteorology on the cloud properties you observed will make for a nice follow-on. I would mention here near the end of this paragraph (or as a separate paragraph as this final paragraph in this section is already quite long) some ideas for how you would investigate the role of meteorology on all these cloud properties. The discussion here, objectively speaking, offers several viable (and accurate) explanations for why you observe these cloud properties, hence quantifying this would be good going forward.

We thank the reviewer for his positive assessment. Our intention was to give a comprehensive overview of the microphysical cloud properties observed during ACTIVATE and to provide a starting point for follow-up studies. In particular, the other instruments onboard the high-flying King Air and the in-situ aerosol measurements provide opportunities for comparisons and detailed process studies.

L369-374: I agree with all of these conclusions. Excellent overview of the in-situ cloud microphysical measurements and observations from ACTIVATE – I definitely learned a lot reading this.

We are pleased with the reviewer's positive feedback.